



# Combining passive- and active-DTS measurements to locate and quantify groundwater discharge into streams

Nataline Simon[1], Olivier Bour[1], Mikaël Faucheux[2], Nicolas Lavenant[1], Hugo Le Lay[2], Ophélie Fovet[2], Zahra Thomas[2] and Laurent Longuevergne[1]

[1]Univ Rennes, CNRS, Géosciences Rennes, UMR 6118, 35000 Rennes, France
[2]UMR SAS, INRAE, Institut Agro, Rennes, France

*Correspondence to*: Nataline Simon (nataline.simon2@gmail.com) and Olivier Bour (olivier.bour@univ-rennes1.fr)

**Abstract.** FO-DTS (Fiber Optic Distributed Temperature Sensing) technology has been widely developed to quantify exchanges between groundwater and surface water during the last decade. In this study, we propose, for the first time, to
combine long-term passive-DTS measurements and active-DTS measurements in order to highlight their respective potential to locate and quantify groundwater discharge into streams. On the one hand, passive-DTS measurements consist in monitoring natural temperature fluctuations to detect and localize groundwater inflows and characterize the temporal pattern of exchanges. Although easy to set up, the quantification of fluxes with this approach often remains difficult since it relies on energy balance models or on the coupling of distributed temperature measurements with additional punctual measurements.
On the other hand, active-DTS methods, recently developed in hydrogeology, consist in continuously monitoring temperature changes induced by a heat source along a FO cable. Recent developments showed that this approach, although more complex to set up than passive-DTS measurements, can address the challenge of quantifying groundwater fluxes and their spatial distribution. Yet it has almost never been conducted in streambed sediments. In this study, both methods are combined by deploying FO cables in the streambed sediments of a first- and second-order stream within a small agricultural
watershed. A numerical model is used to interpret passive-DTS measurements and highlight the temporal and spatial dynamic of groundwater discharge over the annual hydrological cycle. We underline the difficulties and the limitations of deploying a single FO cable to investigate groundwater discharge and show the impact of uncertainty on sediments thermal properties on the quantification of groundwater inflows. On the opposite, the active-DTS experiment allows estimating the spatial distribution of both the thermal conductivity and the groundwater flux at high resolution with very low uncertainties
all along the heated section of FO cable. Our results highlight the added values of conducting active-DTS experiments, eventually combined with passive-DTS measurements, to fully investigate and characterize patterns of groundwater-stream water exchanges at the stream scale. The combination of both methods allows discussing the impact of topography and hydraulic conductivity variations on the variability of groundwater inflows in headwater catchments.





# 1 Introduction

Understanding groundwater and stream water interactions as integral components of a stream catchment continuum is crucial for efficient development and management of water resources (Bencala, 1993; Brunke and Gonser, 1997; Sophocleous, 2002). Particularly essential for the preservation of groundwater dependent ecosystems and riparian habitats (Kalbus et al., 2006), these interactions play a major role on physical, geochemical and biological processes occurring in the stream or in the hyporheic zone (Frei et al., 2019; Jones and Mulholland, 2000). More specifically, these exchanges control

water quality affecting river ecohydrology and hydrochemistry, particularly during dry periods when groundwater is the principal contribution to stream discharge (Brunke and Gonser, 1997). However, localizing and quantifying exchanges between groundwater and stream water is often difficult as these exchange are controlled by multi-scale processes and are therefore highly variable in time and in space (Brunke and Gonser, 1997; Fleckenstein et al., 2006; Flipo et al., 2014; Harvey and Bencala, 1993; Kalbus et al., 2009; Varli and Yilmaz, 2018; Woessner, 2000).

A wide range of methods exists to estimate water fluxes between stream and groundwater including solute tracer concentrations (Brandt et al., 2017; Liao et al., 2021), seepage meter measurements (Rosenberry et al., 2020) or the use of heat as a groundwater tracer (Anderson, 2005; Constantz, 2008), which is particularly efficient in identifying patterns of focused discharge. The approach relies on the detection of temperature anomalies observed at the sediment-water interface (Tyler et al., 2009; Sebok et al., 2013; Westhoff et al., 2011) or into the streambed (Krause et al., 2012; Lowry et al., 2007)

when significant differences exist between groundwater and stream water temperatures. Then, the comparison of temperature variations monitored at different depths in the streambed provides information on groundwater discharge (Anderson, 2005; Constantz, 2008; Hatch et al., 2006; Keery et al., 2007; Lapham, 1989; Stallman, 1965; Webb et al., 2008; Winter et al., 1998). Indeed, the diurnal or seasonal water temperature variations propagates deeper for losing streams (downward conditions) than for gaining streams (upward conditions), since heat transfer is either attenuated or enhanced by groundwater

discharge (Constantz, 2008; Goto et al., 2005). Thus, the use of Vertical Thermal Profiles is widely applied for determining flow directions, quantifying groundwater discharge (Hatch et al., 2006; Lapham, 1989; Keery et al., 2007) and estimating hydraulic parameters (Constantz and Thomas, 1996). However, only a local-scale characterization of the stream-aquifer interactions is achievable with this approach while extensive information on spatial and temporal temperature patterns are required to gain a more complete understanding at reach scale, and even at watershed scale.

This was made possible by the development and the use of the Fiber Optic Distributed Temperature Sensing (FO-DTS) technology for environmental applications (Selker et al., 2006b, a; Shanafield et al., 2018; Tyler et al., 2009). FO-DTS provides continuous temperature data through space and time along fiber optic cables at high spatial resolution (Habel et al., 2009; SEAFOM, 2010; Ukil et al., 2012). By deploying FO cables at the bottom of the stream, the DTS technology allows temperature monitoring of the longitudinal linear stream/sediments interface allowing detecting thermal anomalies induced

by groundwater discharge into the stream (Briggs et al., 2012; Gilmore et al., 2019; Koruk et al., 2020; Moridnejad et al., 2020; Rosenberry et al., 2016; Selker et al., 2006b, a; Westhoff et al., 2007, 2011). This approach was also used to study





seasonal and temporal fluctuations of groundwater discharge into streams (Matheswaran et al., 2014; Slater et al., 2010) and into a lake (Sebok et al., 2013). To go further and in order to avoid signal temperature loss induced by larger river flows, some studies proposed to detect thermal anomalies in the streambed by burying the FO cable into the streambed sediments in

the hyporheic zone (Krause et al., 2012; Le Lay et al., 2019b; Lowry et al., 2007) improving the possibility of localizing groundwater inflows. However, beyond the localization of inflows, the distributed quantification of fluxes from passive-DTS measurements remains even more challenging. It often requires additional localized fluxes estimates or measurements (Briggs et al., 2012; Koruk et al., 2020; Krause et al., 2012; Rosenberry et al., 2016). Otherwise, energy balance models (Selker et al., 2006b; Westhoff et al., 2011) require monitoring significant temperature changes over time that often limits

the application of the method to large groundwater inflows or small headwater streams. The implementation of distributed vertical temperature profiling as proposed by Mamer and Lowry (2013) remains particularly difficult to apply in the field. It requires coupling distributed temperature measurements with punctual vertical temperature profiles (Le Lay et al., 2019a), which raises the question of the extrapolation of punctual measurements to the whole profile. It also raises the question of the estimation of associated uncertainties.

75    To address the difficulty of using natural temperature variations for estimating fluxes, active-DTS methods have been recently developed for different environmental applications. Contrary to passive-DTS methods, active-DTS methods, consisting in injecting a heat source in the sediments, better constrain the signature of diffusion and advection along a FO cable (Bense et al., 2016). Indeed, the difference of temperature measured between an electrically heated and a non-heated FO cable is directly dependent on water fluxes (Bense et al., 2016; Read et al., 2014; Sayde et al., 2015), offering the

possibility to determine fluxes and their spatial distribution over a large range with an excellent accuracy (Simon et al., 2021). Active heat tracer experiments using fiber-optic DTS have been used to estimate wind speed in the low atmosphere (Lapo et al., 2020; Sayde et al., 2015), in dam monitoring (Ghafoori et al., 2020; Perzlmaier et al., 2004; Su et al., 2017), for groundwater fluxes measurements in open (Banks et al., 2014; Klepikova et al., 2018; Read et al., 2014, 2015) and sealed boreholes (Munn et al., 2020; Selker and Selker, 2018) or else in direct contact within sedimentary aquifers (del Val et al.,

2021; des Tombe et al., 2019). Despite promising developments, active-DTS methods have been seldom used in hydrology to estimate groundwater/surface water interactions. High-resolution vertical temperature profiling has been used to quantify vertical fluxes but only permits a local-scale characterization of the stream-aquifer dynamic (Briggs et al., 2016). Although Kurth et al., (2015) coupled passive and active-DTS measurements and highlighted areas with lower and higher flow rates over the cable, the quantification of fluxes remains unexplored.

90    In this context, we propose in this study to couple and compare long-term passive-DTS measurements with active-DTS measurements, which has never been achieved until now. In particular, we aim to discuss the advantages and limitations of high-resolution DTS measurements to locate and quantify groundwater discharge into streams. For doing so, FO cables were deployed in the streambed sediments of a headwater stream within a small agricultural watershed. In the following, we first present the headwater watershed and the experimental setup before presenting the methods used to



interpret both passive- and active-DTS measurements. The fluxes estimates obtained with both passive- and active-DTS measurements are then compared and the advantages and limitations of each method are finally discussed.

## 2 Materiel and methods

### 2.1 The experimental setup on the Kerrien watershed

#### 2.1.1 The Kerrien watershed

The experiment has been achieved on the Kerrien watershed located in South-western Brittany (4°7'24.87''O:47°56'26.97''N). It is part of the AgrHys Environmental Research Observatory, whose principal aim is to understand and characterize transit times in small agricultural catchments (https://www6.inra.fr/ore_agrhys). The site is a part of the French network of critical zone observatories (Gaillardet et al., 2018) and supports extensive hydrological and geochemical researches. Consequently, many instruments and equipment are installed on this experimental site, as detailed
in Fovet et al. (2018), which is an advantage to test new methods and measurement tools in the field.

As shown in Fig. 1, the watershed is a headwater watershed with a second-order stream, subdivided in three first-order sub-watersheds namely the Kerrien, the Kerbernez and the Gerveur sub-watersheds. The Kerrien sub-watershed is a small agricultural watershed (9.5 ha) with higher slopes in the upper parts (14%) than in the bottom lands (5%), where a large wetland was developed (Ruiz et al., 2002). As pointed out in Fig. 1, downstream the wetland, the fields were converted
into a golf course. This is a man-made environment where the stream has been completely restored and dammed to facilitate maintenance. Drainage pipes contribute to drain precipitation from the watershed area directly into the stream, limiting the potential groundwater recharge by draining precipitation from the watershed area into the stream. Further downstream, the stream reaches a natural wood plain.





**Figure 1: Description of the watershed with the location of piezometers, gauging station and fiber optic cables.**

### 2.1.2 Hydrological dynamics of the study site

The Kerrien watershed has been particularly studied and instrumented for estimating transit times in a small agricultural watershed (Fovet et al., 2015a; Martin, 2003), as shown in Fig. 1. For this study, we are using the data from the piezometer transfect F (Fig. 1) including the hillslope piezometer F5b (20 m depth) and the mid-slope piezometer F4 (15 m depth) as markers for the deep groundwater storage dynamics and the riparian piezometers F2 (2 m depth) and F1b (5 m depth) as markers for the riparian groundwater storage dynamics. The gauging station E30 provides stream flow rate.

Previous studies demonstrated that runoff is negligible, so that most of effective precipitations are infiltrating in this headwater watershed. The annual rainfall (1114 mm in average) is well-distributed over the year but recharge mainly occurs in autumn and winter. Therefore, the contribution of groundwater to the stream flow reaches 80-90% (Fovet et al., 2015b; Martin et al., 2006; Ruiz et al., 2002) with the stream discharge during high water periods being highly correlated with





hillslope head gradient (Martin, 2003). As shown in Fig. 2a, piezometric levels show clear seasonal fluctuations with high

levels during winter and spring and low levels during summer and autumn. The hydraulic gradient between the aquifer and

the stream as well as the evolution of the stream discharge suggest that groundwater discharge into the stream should be

particularly expected during the high water level period (From December to June).

**Figure 2: a. Changes in stream flow and in piezometric levels along the transect F over three years. b. Stream and groundwater temperature fluctuations over time along the transect F. The red-colored area corresponds to the period of passive-DTS measurements, conducted from December 2015 to the July 15th, 2016.**

135       Figure 2b shows temperature fluctuations in the stream and in piezometers over a time period from July 2013 to

May 2017. While the groundwater temperature is almost constant in the upslope domain (piezometers F5b and F4),

temperature variations recorded in the stream and in the downslope domain (F2 and F1b) show larger variations following





daily and seasonal temperature variations. Considering the relatively small difference of temperature between groundwater and stream water, burying the FO cable within the sediments should facilitate the detection of potential temperature

anomalies as marker of groundwater discharge (Krause et al., 2012; Le Lay et al., 2019b; Lowry et al., 2007). Otherwise, active-DTS measurements should highlight advective heat transfer, controlled by groundwater discharge.

## 2.2 Passive DTS measurements and data interpretation

### 2.2.1 FO cable deployment and data acquisition

The passive-DTS experiment corresponding to the long-term monitoring of temperature in the streambed was

achieved from December 2015 to July 2016 in the southern part of the study site, as shown in Fig. 1. The FO cable has been deployed from the Kerrien spring and in total, more than 530 m length of BruSens FO cable has been buried directly into the streambed. Due to some obstacles (coarse gravels, cobbles, gauging stations, etc.), it was not possible to bury the FO cable in few places. Everywhere else, the burial depth was estimated in average at 8 cm. The first 165 meters of the FO cable have been deployed in the Kerrien sub-watershed, where the stream is surrounded by a wetland. The streambed is formed by sand

and sludge whose thickness is low but large enough to bury properly the cable. Then, besides a harder substrate, the FO cable was deployed in the golf area. In few local places, the burying was not possible and FO cable was set on the streambed. The last 70 meters of the FO cable have been deployed in a wood plain, a natural environment, where a thick sandy riverbed facilitates cable burying. As highlighted in Fig. 2a, the nine-month experiment insured the monitoring of streambed temperature during both high and low water tables. The highest levels were recorded from January to mid-April, period over

which the wetland is saturated up to the surface.

The FO cable has been connected to a FO-DTS control unit, a Silixa XT-DTS instrument (5 km range). The DTS unit was configured in double-ended configuration (van de Giesen et al., 2012) to collect data at 25 cm and 10 minutes sampling interval. Two calibration baths (one at the ambient temperature and a fridge), as well as PT100 probes (0.1°C) and RBR SoloT probes (0.002°C accuracy) were used to calibrate the data. To assess the accuracy of temperature measurements,

a RBR SoloT probe was set up at the gauging station E30 located at the entrance of the wood. Comparison between DTS measurements and RBR SoloT probes validated the temperature measurements, with a relative uncertainty of measurements estimated at 0.05°C and absolute uncertainty of measurement that can reach at maximum estimated at 0.2°C depending on the period of measurement.

In complement, 4 vertical temperature profiles (VTP) were installed in the streambed in the wetland area by

deploying temperature sensors at 12.5 and 22 cm-depth in the streambed sediments. The position of these punctual sensors in the stream is shown in Fig. 1 and was chosen at locations where groundwater discharge has been observed using preliminary results obtained from FO-DTS monitoring. From upstream to downstream, the VTP are numbered from 1 to 4. For each location, the evolution of temperature was recorded from April 07[th], 2016 to May 03[rd], 2017 using HOBO U12-015-02



sensors (±0.25°C precision). These VTP will be used to quantify local groundwater discharge and results will be compared
with estimates from FO-DTS measurements.

### 2.2.2 Data interpretation

The localization of groundwater inflows can be done by localizing temperature anomalies. Those can be easily
identified by plotting the evolution of the temperature over time and space, especially for long time series. Thermal
anomalies can also be identified using an analysis of the standard deviation of temperature for a given period (Sebok et al.,
2015). Indeed, the calculation of the standard deviation provides insights about the thermal inertia mostly linked to
groundwater fluxes towards the surface and can therefore be used to highlight relative variations of fluxes along the cable.

Then, to quantify vertical fluxes, we use the FLUX-BOT model, a code proposed by Munz and Schmidt (2017),
using a numerical heat transport model to solve the 1D heat transport equation (Carslaw and Jaeger, 1959; Domenico and
Schwartz, 1998). This 1D model aims to calculate vertical fluxes by inversing measured time series observed at least at three
different depths. Temperature variations are simulated according to the optimized fluxes. The quality criteria calculated
between the simulated temperatures and the measured one (NSE, R² and RMSE) allow discussing the quality of flux
estimates. Thus, the model estimates the direction and the intensity of the flow and may highlight the temporal variability of
exchanges.

To apply the model, the stream temperature and the groundwater temperature measured in the piezometer F4,
steady over time (Fig. 2b), were chosen as upper and lower boundary conditions respectively. The stream temperature was
measured for the wetland area at the Kerrien spring with a punctual temperature sensor and at the gauging station E30 (RBR
SoloT) for the wood plain area. The stream temperature is assumed homogeneous for each area since shade and lighting
conditions and water depths are similar. Moreover, the transit time from the spring to the gauging station is relatively rapid
due to the short distances and the river slopes. The FLUX-BOT model was first used for interpreting the temperature
measurements of the four VTP installed in the streambed. Considering the upper and lower boundary conditions previously
defined, the numerical model reproduces the temperature evolution collected at 12.5- and 22-cm depth and provides an
estimate of vertical fluxes for each profile. Details about the interpretation of VTP using the FLUX-BOT model are provided
as supplement (Fig. S1 and S2). The results are compared in the section 3.3 with estimated fluxes from passive- and active-
DTS measurements. Identically, the FLUX-BOT model is used to reproduce and interpret passive-DTS measurements
collected at various spots along the cable. A loop was added in the initial code allowing the interpretation of data collected
for each measurement point. For both applications, the vertical mesh size of the model was set at 0.01 m, as recommended
by Munz and Schmidt (2017). Concerning the thermal properties of the saturated sediments, the volumetric heat capacity
was set at $3 \times 10^6$ J.m$^{-3}$.K$^{-1}$ and various values of thermal conductivities, ranging between 0.9 and 4 W.m$^{-1}$.K$^{-1}$, were tested.





### 2.3 Active DTS measurements

**2.3.1 FO cable deployment and data acquisition**

The first 3 months of passive-DTS measurements showed clear and significant temperature anomalies along the cable, especially in the wetland area, interpreted as potential groundwater exfiltration zones (See the results section below). Consequently, active-DTS measurements were conducted in April 2016 concurrently with passive-DTS measurements by deploying an additional FO cable in the streambed in the wetland, as shown in Fig. 1.

Figure 3 presents the experimental setup of the active-experiment. A FO cable is electrically heated through its steel armoring and the elevation in temperature, associated to the heat injection, is continuously monitored all along the heated section using the FO inside the cable. Without any flow, heat transfers occur through the porous media only by conduction (Constantz, 2008) and a gradual and continuous increase of temperature is therefore expected (Simon et al., 2021). If water flows through the porous medium, advection partly controls the thermal response by dissipating the heat produced by the

heat source. The higher the water flow, the lower should be the temperature increase (Simon et al., 2021). Thus, contrary to passive-DTS measurements, groundwater fluxes can be investigated in any conditions, independently of natural temperature gradients. However, the same hypothesis is assumed about 1D flow considered as perpendicular to FO cables (Simon et al., 2021).

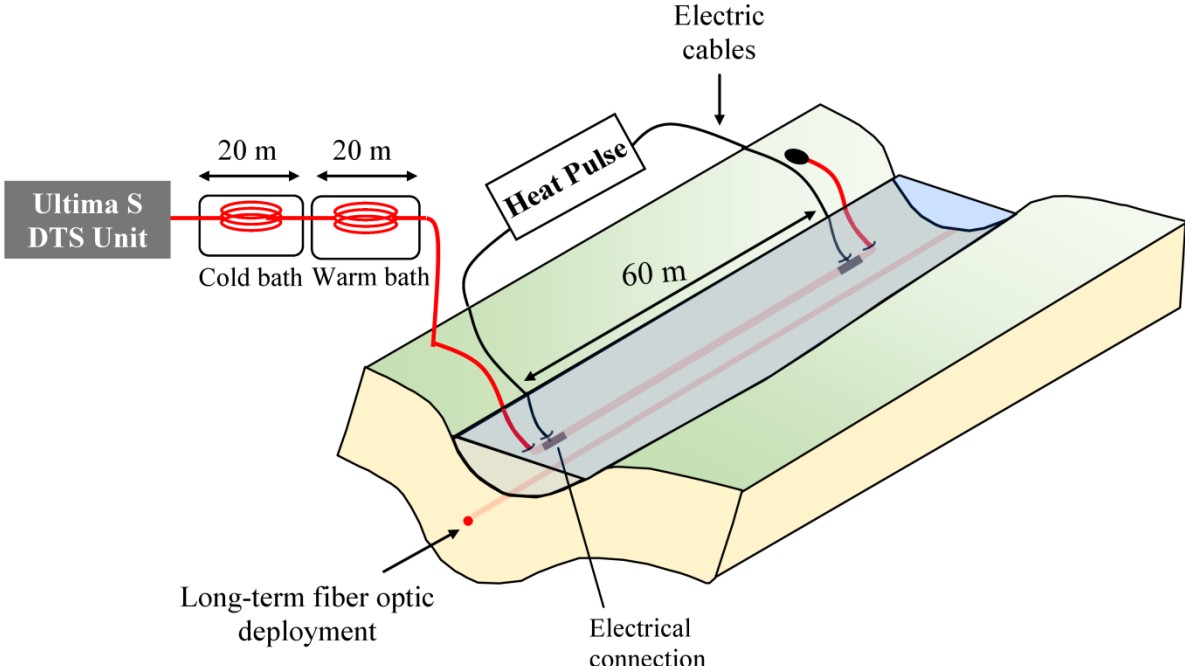

**Figure 3: Experimental setup of the active experiment: a 60 meter-section of a heatable cable has been electrically isolated, buried in the sediments and then heated by connecting to a power controller.**





A 150 m length of BruSens FO cable has been connected to a FO-DTS control unit, a Silixa Ultima S instrument. The unit was configured in double-ended configuration to collect data at 12.5 sampling cm and 60 seconds time interval. The effective spatial resolution of DTS measurements with this unit was estimated varying between 66 and 90 cm following the methodology proposed in Simon et al. (2020). The calibration process applied was similar to the one applied for calibrating passive-DTS measurements.

A 60-meters section of the 150 m FO cable has been buried in the streambed in parallel with the FO cable previously installed for the passive experiment. As the heating experiment induces a very localized thermal perturbation within the streambed, the non-heated cable was not affected by the heat injection. Thus, natural streambed temperature fluctuations were monitored during the experiment using the non-heated cable. It should be noted that, during the FO cable deployment in the streambed, local heterogeneities led to the impossibility to deploy the whole FO heating section in the streambed. Thanks to cable numbering, these non-buried sections were accurately located. For buried sections, the burial depth was measured in situ and estimated to be around 8 to 10 cm. This 60-m section has been electrically isolated and heated using a power controller (provided by CTEMP, https://ctemps.org/) supplying a constant and uniform heating rate power of 35 W.m$^{-1}$ along the heated FO cable. The heated cable has been energized continuously during 4 hours and the recovery was also monitored for an additional 3 hours after turning off the power controller.

### 2.3.3 Data interpretation

Here, we interpreted the data using the ADTS Toolbox proposed by Simon and Bour (Submitted) for automatically interpreting active-DTS measurements. The ADTS Toolbox contains several MATLAB codes that allow estimating the thermal conductivities and the groundwater fluxes and their respective spatial distribution all along the heated section. It uses for the data interpretation an analytical approach proposed and validated by Simon et al. (2021), that consists in defining, for each measurement point along the heated section, the optimized values of thermal conductivity and flux that allow reproducing at best the associated temperature increase measured over the heating period. The use of the ADTS toolbox also provides an estimate of the associated uncertainties (Simon and Bour, Submitted).

## 3 Results

### 3.1 Passive-DTS measurements

### 3.1.1 Spatial variability of temperature signals

Figure 4a synthesizes the results of the passive-DTS experiment and shows temperature signals monitored all along the FO cable deployed in the streambed sediments. The x-axis indicates the distance between the Kerrien spring (located at 0 m) and each measurement point in the streambed. Temperature variations are presented from December 2015 to July 2016 (y-axis). In June and July, despite very low flows, the stream never dried up. Two different behaviours are highlighted in the



figure. On the one hand, vertical yellow lines can be observed near the Kerrien Spring in the first 150 meters of cable. These lines emphasize that the temperature recorded in these areas is relatively constant over time (few temperature variations are recorded). On the other hand, away from the spring, beyond 300 m, clear and large differences in temperature are observed

250 between colder periods (from December to mid-April) and warmer periods (from mid-April to July).

**Figure 4: a. Long-term monitoring of streambed temperature along the river using DTS. Sections S1, S2 and S3 match with spots where the cable lies on the bank because of obstacles in the stream (gauging stations). Temporary thermal anomalies located for instance in 425 m and around 500 m correspond to air-exposed periods during which the cable was not held at the sediment/water**
255 **interface. b. Standard Deviation (SD) of the temperature calculated over the experiment duration for each measurement point along the FO cable. Sections where the cable was outside the stream or punctually unburied were removed. The red line represents the SD of the stream temperature (1.38°C) measured at the gauging station E30.**





To highlight the spatial variability of the temperature signal, the Standard Deviation (SD) of temperature was calculated for each measurement point over the whole duration of the experiment as presented in Fig. 4b. In the first 50-m of measurements, near the spring, the value of the SD of temperature is relatively stable and very low (around 0.37 °C). Then, it progressively increases from upstream to downstream and stabilizes around the value of the SD of the stream temperature (≈ 1.375 °C) at around 300 meters (in the middle of the golf area). Interestingly, the value of the SD directly reflects the amplitude of daily temperature fluctuations.

The lowest SD values are recorded in the upstream wetland (d < 160 m in Fig. 4b). In this area, as illustrated in Fig. 5a by the red curve (94.56 m, σ = 0.2 °C), the temperature is relatively stable over time (around 12.5°C) and the daily stream temperature fluctuations are widely attenuated. However, significant differences are observed between temperature measurements from upstream to downstream, as highlighted by the progressive increase of SD measured from the spring to 160 m. This increase reflects the increase of the amplitudes of daily temperature fluctuations collected from upstream (orange line in Fig. 5b, σ = 0.36 °C) to downstream (grey line in Fig. 5b, σ = 0.95 °C). In addition, the profile of SD (Fig. 4b) also shows punctual "peaks" associated to very low SD values, in agreement with the yellow lines observed in Fig. 4a. These peaks can be associated to spots where the amplitude of temperature is low all over the period of measurements, as illustrated by blue curve in Fig. 5b (53.38 m), where the value of the SD is equal to 0.24 °C. As we shall see in the next sections, the relative temperature stability suggests that these peaks or "hotspots" may be associated to local groundwater discharge.

Further downstream (from 220 m up to 300 m in the first part of the golf area), while the value of the SD progressively increases (Fig. 4b), higher amplitudes of daily temperature variations are monitored as illustrated in the Fig. 5a by the blue curve (227.5 m, σ = 0.93 °C). In this area, SDs values are lower than the one calculated in the stream (1.3 °C, Fig. 4b) meaning that the daily temperature fluctuations are attenuated. Once again, the progressive increase of the value of the SD highlights differences in temperature amplitude (the further, the higher the amplitudes of temperature). Finally, in the second part of the golf area and in the wood (starting from approximately 300 m), SDs values tend towards the value of the SD of the stream (1.3 °C, Fig. 4b). The associated streambed temperature variations are almost identical to the stream variations, as illustrated in the Fig. 5a by the yellow line (357.91 m, σ = 1.42 °C). Note here that the SD evolution shows a well-marked step at 300 m from 1.2 to 1.4°C, exactly at the confluence between the Kerrien and the Gerveur streams (see Fig. 1). Moreover, very punctual decreases of SDs can be observed between 402 and 425 m, where significant thermal anomalies are monitored from mid-February to mid-April.







**Figure 5: a. Examples of streambed temperature variations measured with the FO-DTS a. at 94.56 m, 227.5 m and 357.91 m from upstream with respective SD values equals to 0.2 °C, 0.93 °C and 1.42 °C; b. in the wetland area at 11.44 m, 53.38 m and 137.27 m from upstream with respective SD values equals to 0.36 °C, 0.24 °C and 0.95 °C.**

290    Streambed temperature measurements clearly show a general trend with an increase of the amplitudes of temperature variations measured from upstream (the spring) to downstream, up to around 300 m. In the first 300 meters,



temperature fluctuations appear attenuated compared to the daily temperature fluctuations and the streambed temperature variations measured more downstream (beyond 300 m). Thus, lower temperature amplitude variations suggest groundwater inflows, especially for the measurement points where the lowest values of SD are recorded (minimal peaks of SD digressing from the general trend, as illustrated by the blue line in Fig. 5b). Indeed, for these "hotspots", thermal anomalies are clearly recorded and the temperature is relatively stable over time according to the stable groundwater temperature (around 12.5-13°C). The general increase of SD from the spring up to 300 m may be associated to a global decrease of groundwater inflows from upstream to downstream in the first 300 meters of the watershed. Higher and punctual inflows would be located in spots where the value of the SD is clearly lower than the general trend. Nevertheless, the gradual increase of the SD could also be due to the fact that the stream water temperature, equal to the groundwater temperature at the spring, may progressively equilibrate with the air temperature when travelling along the stream.

To summarize, hotspots associated to minimal peaks of SD are certainly associated to groundwater discharges, but the general evolution of temperature SD may be due to different factors.

### 3.1.2 Temporal variability of temperature signals

Three different time periods were clearly identified with passive-DTS measurements according to the behaviour of the streambed temperature evolution over time: P1, P2 and P3 (Fig. 4 and 5). In order to characterize the temporal dynamic of groundwater discharge, the daily SD of temperature was calculated along the FO cable for each day of measurement as shown in Fig. 6. During the first month of monitoring (period P1 in Fig. 4a), quite similar temperature variations are observed all along the cable independently of the localization of the measurement point over the watershed. The temperature fluctuates between 9 and 14°C according to daily temperature fluctuations, as illustrated in Fig. 5a. The red line in the Fig. 6 is a representative example of the daily SD profiles calculated in December. During this period, the value of the daily SD calculated in the wetland area, is higher than the daily SD of the stream temperature (horizontal red line) and also higher than the daily SDs calculated in the golf or in the wood areas. Note also the piezometric levels measured near the stream during P1 are lower than the stage level and very low flow rate are measured in the stream in December (mean ≈ 2 L.s$^{-1}$) (Fig. 2a).

The second period from January to mid-April (period P2 in Fig. 4a) is characterized by well-marked temperature anomalies observed in the wetland area, where the streambed temperature stabilizes around 12.5°C (red line in Fig. 5a). During this period, the longitudinal profiles of daily SD of temperature, represented by the green line in Fig. 6, are identical to the one calculated over the experiment duration (Fig. 4b). This period is also marked by an increase of the piezometric level and water stage (Fig. 2a). Recharge starts end of December and the elevation of the groundwater table becomes higher than the elevation of the stream stage early January.

Lastly, from mid-April to the end of the monitoring (period P3 in Fig. 4a), the mean streambed temperature measured all along the cable and the daily temperature amplitudes both increase. However, the increase of the daily temperature amplitude, and thus of the SD, is more important in the downstream section (> 300m), as highlighted in Fig. 5a. Contrary to temperature measurements recorded during P1 (December), the temperature variations measured during P3 in





upstream sections (red and blue lines in Fig. 5a) are very different of the one measured downstream (orange line in Fig. 5a).
Note also that the piezometric level measured during P3 gradually decreases as well as the stream flow (Fig. 2a). According
to Fig. 6, the behaviour of the temperature SD is clearly different from each period of time. This suggests therefore that the
temperature SD may be a good marker of possible groundwater inflows although it provides only qualitative information.

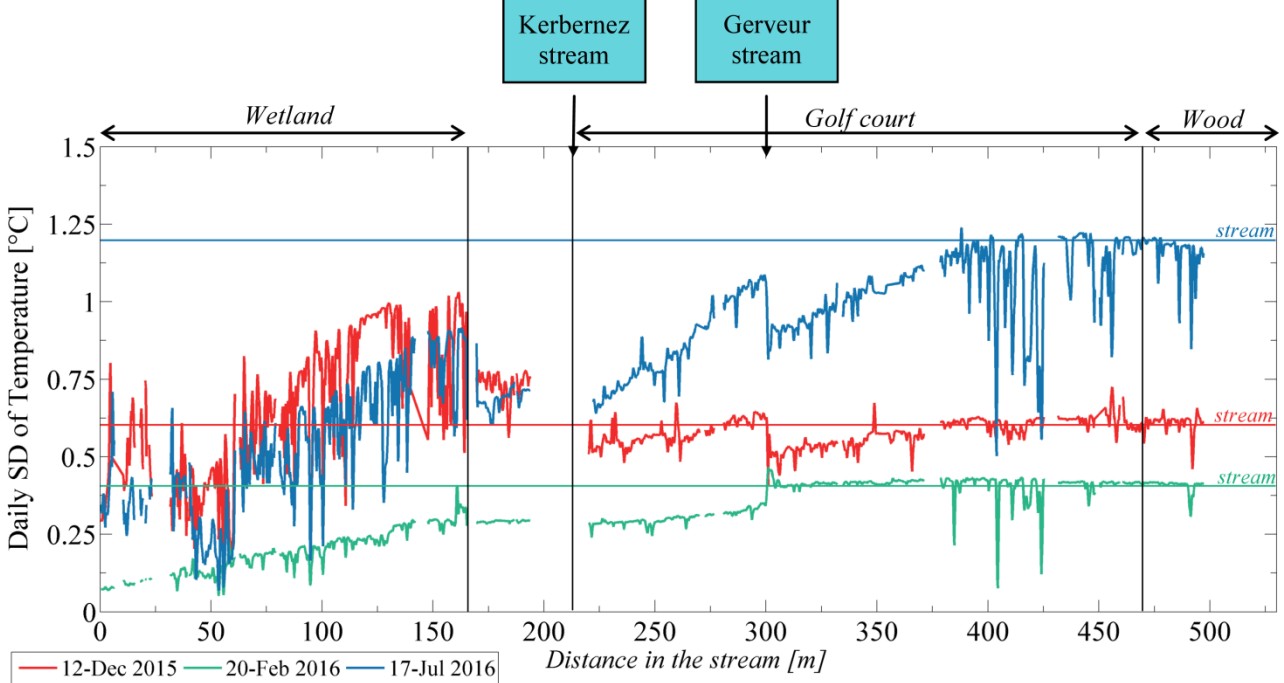

**Figure 6: Examples of longitudinal profiles of daily SD of temperature calculated the Dec. 12, 2015 (red line), the Feb. 20, 2016 (green line) and the Jul. 17, 2016 (blue line). These profiles are representative of daily SD profiles calculated during the periods P1, P2 and P3 respectively. The coloured horizontal lines correspond to the daily SD of stream temperature calculated the same days.**

### 3.1.3 Quantifying groundwater/ stream water exchanges

To go further and interpret streambed temperature variations, the FLUX-BOT model was applied for each
measurement point. A detailed example of the application of the FLUX-BOT model on a single measurement point
(d = 5.08 m) including the quality criteria associated to the fluxes estimates is presented in Fig. S3. Although the model was
applied for each measurement point, simulated temperature variations are consistent only with DTS measurements located in
the first 150 meters of the temperature profile (SD < 1°C) with values of NSE > 0.74, values of R² > 0.85 and values of
RMSEs < 0.9°C. Beyond 150 m, the quality of the results considerably decreases with values of NSE < 0.6, values of
R² < 0.65 and values of RMSEs > 1.8°C. Thus, the uncertainties on fluxes estimates are too large in this lower part of the
watershed (for d>150 m) to estimate groundwater discharge. Consequently, the model is found not applicable to interpret
temperature measurements and results are not provided here.



Figure 7 shows the results of the application of the FLUX-BOT model on passive-DTS measurements collected along the cable deployed in streambed sediments in the wetland area, for d < 150m, where the model is applicable. The
model predicts negative values of fluxes all along the interpreted section, indicating upward water flow, which strengthens the assumption of groundwater discharge into stream. However, as shown in Fig. 7b, groundwater fluxes estimates are strongly dependent on the value of thermal conductivity of the sediments used in the model. Indeed, the model was applied for 3 values of thermal conductivity (1, 2.5 and 4 $W.m^{-1}.K^{-1}$). The streambed sediments being composed of saturated clay, silt, sand and gravel, the thermal conductivity may typically range between 0.9 and 4 $W.m^{-1}.K^{-1}$ (Stauffer et al., 2013). By
varying the thermal conductivity from 1 $W.m^{-1}.K^{-1}$ (blue line) to 4 $W.m^{-1}.K^{-1}$ (green line), the estimated discharge is around 4 times higher. For instance, at d = 75 m, the mean flux is estimated $-3.43 \times 10^{-6}$ $m.s^{-1}$ for $\lambda=1$ $W.m^{-1}.K^{-1}$ against $-1.48 \times 10^{-5}$ $m.s^{-1}$ for $\lambda = 4$ $W.m^{-1}.K^{-1}$. Note however that the results are more sensitive to the value of the thermal conductivity when groundwater inflows are higher (see Fig. S3 for more details).

Whatever the uncertainties, higher groundwater inflows are estimated upstream, at the head of the watershed and
close to the spring (Fig. 7b). The comparison with the SD profile (Fig. 7a) tends to confirm the correlation between the value of the SD of streambed temperature and the importance of groundwater discharge. The increase of SD from upstream to downstream seems associated to a decrease of groundwater discharge. The inflows are estimated twice higher near the spring than downstream. Hence, assuming a thermal conductivity of the sediments $\lambda = 2.5$ $W.m^{-1}.K^{-1}$ (orange line in Fig. 7b), the mean flux is estimated $-1.24 \times 10^{-5}$ $m.s^{-1}$ near the spring (d = 0 m) while it only reaches $-6.55 \times 10^{-6}$ $m.s^{-1}$ for d = 150 m. Results
also suggest that local peaks of SD (at 95 or 100 m for instance) can be associated to punctual preferential pathways, where groundwater discharge is locally estimated four times higher than elsewhere.

Figure 7c shows the temporal variability of groundwater discharge estimated for different measurement points. The range of fluxes is larger from January to May with groundwater inflows varying between $5 \times 10^{-6}$ and $2.5 \times 10^{-5}$ $m.s^{-1}$ by considering $\lambda=2.5$ $W.m^{-1}.K^{-1}$. Lower groundwater inflows are detected during the first month of experiment ($<7.5 \times 10^{-6}$ $m.s^{-1}$)
and at the end of the experiment ($<6 \times 10^{-6}$ $m.s^{-1}$). The same temporal dynamic is observed for all data collected in the wetland area.



**Figure 7: a. Profile of SD of the streambed temperature calculated over the experiment duration for each measurement point located along the FO cable and deployed in the wetland area; b. Profiles of mean fluxes estimated over the experiment duration using the FLUX-BOT model from DTS measurements collected in streambed sediments in the wetland area considering 3 values of thermal conductivity (Negative values indicate upward water flux); c. Temporal evolution of the estimated flow considering $\lambda$=2.5 W.m$^{-1}$.K$^{-1}$.**





### 3.2 Active-DTS measurements

**3.2.1 Data interpretation**

Among active-DTS measurements, 172 measurements points, selected following the data processing method developed in Simon et al. (2020), are thought to be significant. Note that the raw data of active-DTS measurements are presented in the Fig. S4 and that the data processing (sorting and quality check) and the definition of significant points are presented in detail in the supplement. Hereafter, we focus on data interpretation and fluxes estimates of the selected

significant data points.

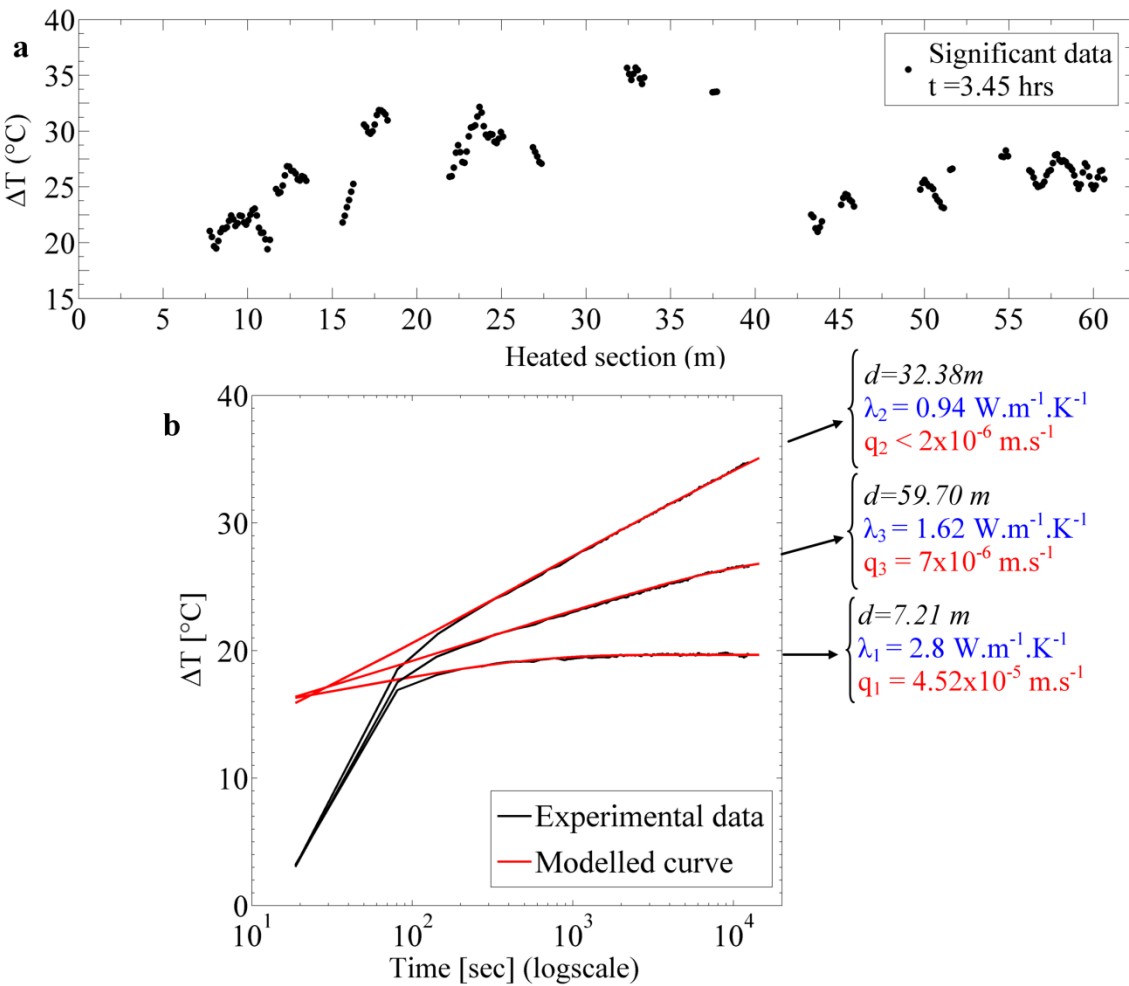

**Figure 8: a. Significant $\Delta T$ values measured 3.45hrs after the start of the heating period measured along the buried FO section. b. Examples of data interpretation on three thermal response curves observed in the data (black lines). The MILS model was used to reproduce the temperature increase during both the conduction- and advection-dominant periods (red lines).**

Figure 8a shows the variability of $\Delta T$ measured 3.45hrs after the start of the heat experiment. Despite some areas without data, due to noise in the data or imperfect burying, the measurement points are distributed over less than 55 m



offering a clear view of the variability of thermal responses. The value of $\Delta T$ is particularly variable and ranges between 19.42 °C (at 11.2 m) and 36°C (at 132.5 m). However, despite the variability observed, adjacent points present in general a similar dynamic with similar values of $\Delta T$, suggesting similar behaviours over a certain range or scale.

Each validated data point was interpreted using the ADTS Toolbox (Simon and Bour, Submitted) to estimate thermal conductivities and fluxes. Figure 8b shows three examples of thermal response curves observed in the data collected along the heated FO section (black lines) and their respective interpretation with the ADTS Toolbox (red lines). The data interpretation focuses on the interpretation of the second part of the temperature increase (for $t > 90$ sec) corresponding to the temperature increase controlled by heat conduction and advection in the sediments (Simon et al., 2021). As illustrated in Fig.

8b, the thermal conductivity highly controls the thermal response and the variability of $\Delta T$. For instance, at $d = 32.38$ m, the temperature rise reaches 34.7 °C after 3.45 hrs of heating, in concordance with the very low thermal conductivity estimated (0.94 W.m$^{-1}$.K$^{-1}$). On the opposite, at $d = 7.21$ m, where the temperature rise is much lower and reaches only 19.7 °C, $\lambda$ is estimated at 2.8 W.m$^{-1}$.K$^{-1}$. The fluxes are then estimated using the temperature at later times, as the intensity of the flux controls temperature stabilization (Simon et al., 2021). Thus, for instance, the following fluxes values of $7 \times 10^{-6}$ m.s$^{-1}$ and

$4.52 \times 10^{-5}$ m.s$^{-1}$ are estimated at $d = 59.70$ m and $d = 7.21$ m respectively. However, for some points, as illustrated by the temperature evolution measured at $d = 32.38$ m, the temperature does not stabilize for later times and keeps increasing with no inflexion over the whole heating period. This implies either no-flow conditions ($q = 0$ m.s$^{-1}$) or very low-flow conditions which presume that the heat duration was not long enough to reach temperature stabilization (Simon et al., 2021). In these cases, only an estimate of $q_{lim}$ can be provided, which corresponds to the highest value of flow that would induce such

temperature increase. For instance, at $d = 32.38$ m, the flux is estimated to be lower than $2 \times 10^{-6}$ m.s$^{-1}$.

### 3.2.2 Spatial variability of thermal conductivities and water fluxes estimates

Figure 9 shows the estimation of both the thermal conductivities (Fig. 9a) and the fluxes (Fig. 9b) obtained from active-DTS measurements using the ADTS Toolbox. It provides an estimate of their respective spatial distribution at very small scale. As shown in Fig. 9a, the thermal conductivities estimated along the heated section vary between 0.8 and 3.14

W.m$^{-1}$.K$^{-1}$, with a median value at 1.65 W.m$^{-1}$.K$^{-1}$. The RMSE calculated between observed data and the best-fit model was systematically lower than 0.05 °C. Seeing the data noise (< 0.1 °C), the maximal uncertainty of these estimates is estimated to be ± 0.2 W.m$^{-1}$.K$^{-1}$.

As shown in Fig. 9b, estimated groundwater fluxes vary between $2 \times 10^{-6}$ and $4.74 \times 10^{-5}$ m.s$^{-1}$, with a mean value at 1.34 $\times 10^{-5}$ m.s$^{-1}$ and a SD of 9.18 $\times 10^{-6}$ m.s$^{-1}$. For 9 locations (blue points), only the value of $q_{lim}$ was evaluated since the

departure of the conduction regime towards temperature stabilization was not reached at the end of the heating period. Note that the data interpretation does not provide the flow direction, the temperature increase being identical for upward and downward conditions. Although significant measurements are not available all along the sections, results show a decrease of the flux from upstream to downstream, particularly in the first twenty meters of measurements. At greater distances, fluxes are more diffuse in space, except at few locations, for instance at 43, 50 and 52 m from the start of the heated section where



higher values are observed. Interestingly, very local and high fluxes values, spreading on less than 2 m, can be observed, as
for instance at $d = 10$ m.

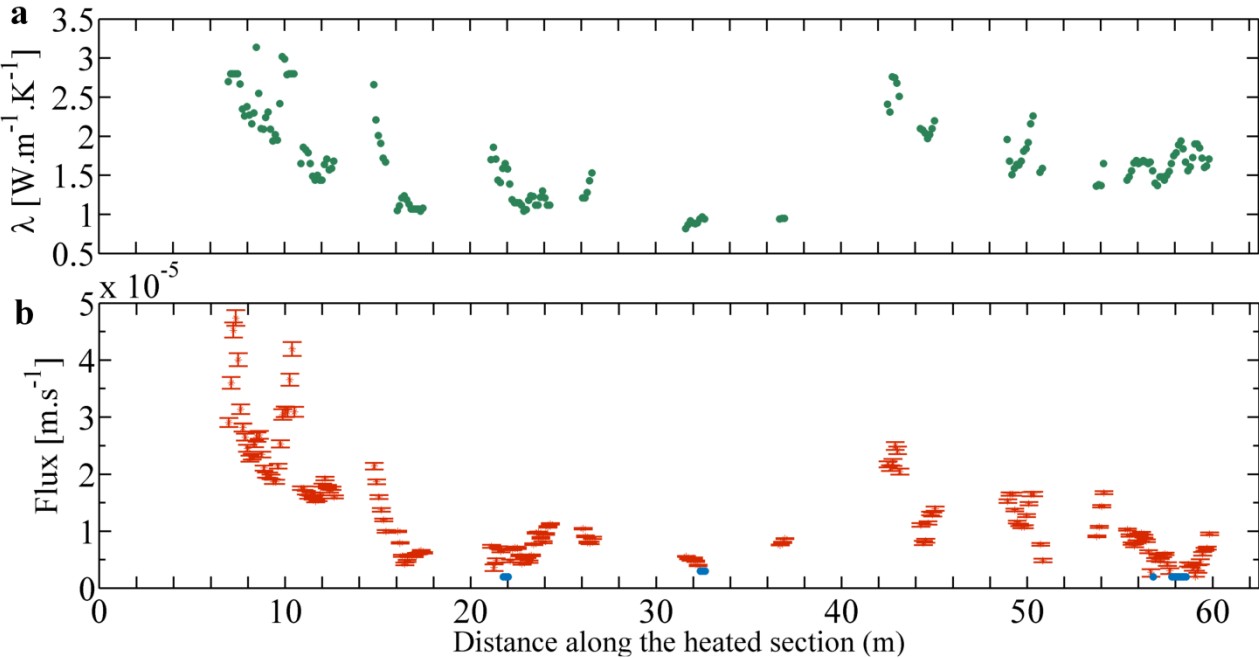

**Figure 9: The interpretation of active-DTS measurements along the heated section of FO cable leads to estimate the spatial distribution of both a. the thermal conductivity and b. the water fluxes and their associated errors (error bars). Blue points mark**
**locations where the temperature stabilization is not reached and where an estimate of $q_{lim}$ is provided. Errors bars corresponding to uncertainties on flow estimates calculated with respect to data noise for each measurement points.**

### 3.3 Comparison between passive- and active-DTS measurements

Figure 10 compares estimated values of groundwater fluxes for the 7th April 2016. The flow direction is assumed
upward in agreement with passive-DTS measurements (Fig. 7b).

For passive-DTS measurements, the two light grey curves correspond to fluxes estimates considering $\lambda = 1$ W.m$^{-1}$.K$^{-1}$ and $\lambda = 4$ W.m$^{-1}$.K$^{-1}$, assuming that fluxes vary between these two thresholds. It appears that the flow quantification
remains uncertain especially because of the lack of knowledge about thermal conductivity variations. Results show a slight
decrease of groundwater discharge from upstream to downstream but the large range of estimated fluxes probably blurs the
actual trend.

Fluxes estimates from active-DTS measurements (pink points) can be qualitatively compared with the evolution of
the SD of temperature (green line). The lowest SD values are located in the first 55 m of the stream, which is in good
agreement with active-DTS measurements that highlighted highest groundwater discharges between 47 and 53 m (between
$1.7\times10^{-5}$ and $4.9\times10^{-5}$ m.s$^{-1}$). Between approximately 55 and 60 m, the value of SD increases rapidly (from 1.25 °C at 54 m
to 2°C at 60 m) while the fluxes estimated from active-DTS measurements decrease linearly (from $2.1\times10^{-5}$ at 56.8 m to





$4.2 \times 10^{-6}$ m.s$^{-1}$ at 58.5 m). From 60 m, the SD increases and the associated values of fluxes estimated from active-DTS are particularly low, varying for instance between $2 \times 10^{-6}$ and $1.1 \times 10^{-5}$ m.s$^{-1}$. Interestingly, the locations of punctual increases of groundwater discharge detected with the active experiment at 87.5 m, 95 m and 100 m (black arrows) match well with punctual decreases of SD values. Note also that flux estimates from active-DTS measurements are in very good agreement with the results of VTPs (blue line). The estimated flux based on passive-DTS measurements at 53.4 m is also in good

agreement with active-DTS results despite the large uncertainty.

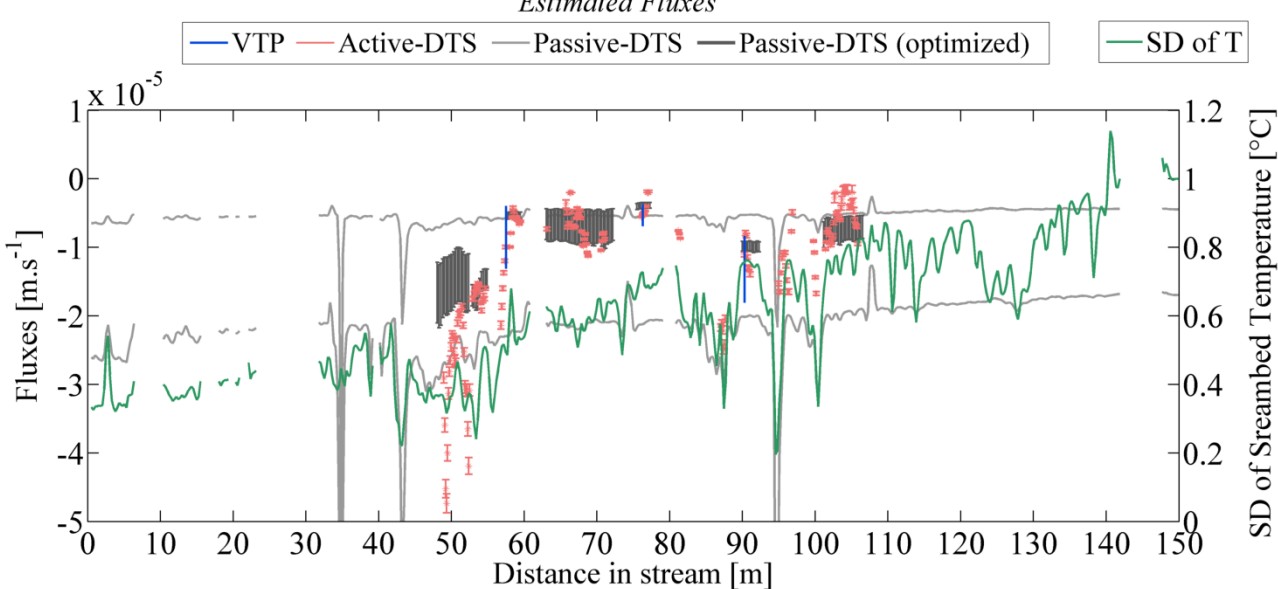

**Figure 10: Comparison of flow estimated the 7$^{th}$ April 2016 using the vertical temperature profiles (VTP), the passive-DTS measurements and the active-DTS measurements. Results are compared with the evolution of SD of the temperature calculated from passive-DTS measurements. The black arrows highlight localized groundwater discharge.**

For passive-DTS measurements, fluxes estimates can be highly improved if the effective value of thermal conductivity is considered in the model. Actually, the data have been once again interpreted with the FLUX-BOT model by considering the thermal conductivity estimated from active-DTS measurements. As a result, the estimated range of fluxes was highly reduced and in much better agreement with other estimates, as shown by dark grey lines in Fig. 10. For instance, between 63 and 72 m, the thermal conductivity was evaluated from active-DTS measurements between 1 and 1.9 W.m$^{-1}$.K$^{-1}$

depending on measurement points (Fig. 9a). Using such values, the fluxes estimated from passive-DTS measurements in this area vary between $-9.5 \times 10^{-6}$ and $-4.3 \times 10^{-5}$ m.s$^{-1}$ (dark grey lines), whereas initially estimated between $-2.2 \times 10^{-5}$ and $-5.4 \times 10^{-6}$ m.s$^{-1}$ considering λ between 1 and 4 W.m$^{-1}$.K$^{-1}$ (light grey lines). As shown in Fig. 10, except between 48 and 52 m where small discrepancies remain, this approach significantly reduces the range of fluxes estimated and shows that passive-DTS results are in good agreement with active-DTS results when an independent and more precise estimate of the thermal

conductivity is considered. Note however that the exact location of the non-heated cable (passive measurements) in relation to the heated cable is difficult to establish precisely which limits the precise location of the thermal conductivity estimates.



## 4 Discussion

### 4.1 Passive-DTS measurements

### 4.1.1 Interpretation of streambed temperature measurements

Burying a FO cable in the streambed allowed recording streambed temperature variations at high resolution during several months along several hundreds of meters. This is clearly the main advantage of passive-DTS method since such achievement would not have been possible with individual temperature probes. Once the FO is installed in the streambed, which can be difficult depending on the streambed nature, the long-term recording of temperature is relatively easy and autonomous.

Streambed temperature measurements recorded with the FO cable allow making first assumptions about the spatial and temporal dynamics occurring in the watershed if used over a several-month-long monitoring. For doing so, the calculation of the Standard Deviation (SD) of streambed temperature over time appears a reliable proxy for locating inflows (Fig. 4 and 5), since higher SDs values can be associated to larger daily temperature amplitudes. Here, the SD of temperature increases progressively from upstream to downstream up to reaching the SD of stream temperature around d = 300 m (Fig.

4b) and very localized and low values of SD, described previously as minimal peaks of SD digressing from the general trend, also appear along the profile. We have already made the assumption that these hotspots could be interpreted as groundwater discharges while the general increase of SD from the spring up to 300 meters could be interpreted either as a general decrease of groundwater discharge from upstream to downstream or as a possible equilibrium between the stream temperature and the atmosphere.

Theoretically, to interpret the data and conclude about groundwater/surface water exchanges, the temperature variations recorded in the streambed should be compared with the stream temperature variations and the groundwater temperature variations (Constantz, 2008). In practice, it would require deploying at least 3 FO cables in the field (Mamer and Lowry, 2013), which seems technically impossible. Therefore, the data interpretation requires assuming the stream temperature homogeneous along the investigated section, which is consistent in our study case since shade and lighting

conditions and water depths are similar. Moreover, the transit time from the spring to the gauging station is relatively rapid due to the short distances and the river slopes and the very low water depths certainly support the fast balance between the air temperature and the stream temperature. This assumption is partly confirmed with the stream temperature recorded near the spring with the DTS, in an area where the FO cable is not buried but lies in the bottom of the stream. At this location, the SD of the temperature equals 1.03°C, which is a little less 1.375 °C, the value recorded in the stream at the gauging station

E30, located at around 490 m of the spring. However, this suggests that the SD of the stream temperature signal should be relatively high, greater than 1°C all along the stream and consequently much larger than the SD recorded within the streambed, especially in the upstream section of the stream (first 300 meters).

    If stream temperature variations are assumed almost uniform along the studied section, the values of SDs inferior to the SD of the stream temperature could actually be interpreted as groundwater inflows. For these lower values, daily





temperature variations are more or less attenuated in proportion to the value of the SD (as shown in Fig. 5). This is strengthened by the fact that, for the measurements points located in the wetland where the lowest values of SD are recorded (minimal 'peaks' of SD illustrated by the blue line in Fig. 5b), clear thermal anomalies are recorded and the temperature appears relatively stable over time according to the stable groundwater temperature (around 12.5-13°C). Therefore, results suggest a decrease of groundwater inflows from upstream to downstream in the first 300 meters of the watershed with higher and punctual inflows in spots where the SD with clearly lower than the general trend.

Beyond 300 m, the values of the SD are almost equal to the SD of the stream temperature and the recorded temperature variations are approximately similar to stream temperature variations. Concerning the second part of the golf area (from 300 to around 470 m), the data interpretation remains difficult because of the hard substrate that limited the burying of the FO cable. However, in the last 70 meters of the FO cable (wood plain), the cable was easily buried in a thick sandy riverbed. Thus, the value of the SD in this area, equals to the one recorded in the stream, would suggest the absence of groundwater inflows, which would remain limited to the wetland and therefore to the upper part of the watershed with highest topographic changes, as we shall discuss later.

**4.1.2 Quantification of groundwater inflows**

To go further, the inverse numerical FLUX-BOT model was used to quantify the vertical fluxes from DTS measurements recorded all along the FO cable. Once again, the stream temperature variations were assumed uniform along the studied section. The model was clearly not applicable in the lower part of the watershed (for d>150 m) because of too large uncertainties on fluxes estimates. Thus, despite the values of the SD recorded between d=150 m and d=300 m, slightly lower than the SD of the stream temperature variations suggesting potential groundwater inflows, the fluxes quantification remains too uncertain in this area to be validated. This is probably because groundwater inflows are too low or diffuse in this part of the watershed.

In the first 150 m, where the model can be successfully used, results firstly allow determining the flow direction, showing that thermal anomalies can be associated to groundwater inflows. This confirms the spatial and temporal dynamics of exchanges occurring in the wetland. However, the numerical model requires making assumptions about boundary conditions or the burial depth of the cable, which appears to be a main limitation for quantifying groundwater inflows. For instance, although the burial depth was estimated around 8 cm, the precise determination of this depth all over the cable was not possible. Complementary tests (not shown here), conducted by varying the depth of the FO cable in the model, suggest that varying the burial depth of ± 2 cm induces in average a difference of ± 50% on fluxes estimates, showing the high uncertainty linked to the burial depth.

Most of all, results showed that thermal conductivity value has an impact on fluxes estimates, which is consistent with the results of Briggs et al. (2014), Duque et al. (2016), Lapham (1989) and Sebok and Müller (2019). The ignorance and assumptions on thermal conductivities values lead to high uncertainties on fluxes estimates using both VTP and passive-DTS measurements. Thus, *in-situ* estimates of thermal conductivities using thermal conductivity probes could considerably



improve the fluxes estimates, as demonstrated by Duque et al. (2016), who reported up to 89% increase in flux estimates when using *in situ* measured sediment thermal conductivities. However, seeing the high spatial variability of the thermal
conductivity highlighted through the active-DTS experiment, it would certainly require a tremendous effort in the field to characterize such variability with single probes.

To summarize, passive-DTS measurements can provide streambed temperature variations at high resolution during several months. As previously demonstrated (Krause et al., 2012; Le Lay et al., 2019b, a; Lowry et al., 2007), this approach can help highlighting possible groundwater inflows. The calculation of the SD is an efficient approach to rapidly identify
strong thermal anomalies (Sebok et al., 2015) and to compare the temperature amplitudes recorded all along the FO cable (Le Lay et al., 2019b). However, the interpretation of temperature measurements and the fluxes quantification depend on strong assumptions about thermal properties and boundary conditions, and especially about the stream temperature. This suggests that burying a single FO cable in the streambed, although very promising and interesting, is not enough to fully characterize groundwater/surface water interactions. For further applications, results suggest the necessity of deploying an
additional FO cable at the bottom of the stream in order to measure also stream temperature variations at high resolution all along the studied section. This seems the only way to fully and efficiently extend the heat-based classical methods (Constantz, 2008; Hatch et al., 2006) at high spatial resolution. A third buried FO cable would be the optimal configuration to estimate distributed vertical fluxes (Mamer and Lowry, 2013) and reduce the uncertainties on fluxes estimations.

## 4.2 Active-DTS measurements

Contrary to passive-DTS measurements, the active-DTS measurements turned out to be an efficient method to estimate fluxes at high resolution and with a very low uncertainty. From only 4 hrs of measurements, the approach provides an estimate of both the thermal conductivities and the fluxes along the heated cable confirming the high variability of these two parameters in space (Fig. 9). It clearly appears that the main advantage of the approach is the low uncertainty on fluxes estimates. Note that the use of the ADTS Toolbox, that automatically interprets active-DTS measurements (Simon and Bour,
Submitted), highly facilitates the data interpretation, which is finally much easier than the interpretation of passive-DTS measurements.

Results show that streambed thermal conductivities are relatively variable in space but consistent with streambed sediments, composed of saturated clay and silt, and saturated sand and gravel, whose thermal conductivity values commonly range respectively between 0.9 and 4 $W.m^{-1}.K^{-1}$ (Stauffer et al., 2013). The large range of thermal conductivities observed in
this relatively small section of streambed (less than 60m) demonstrates the interest of distributed measurements to fully characterize the streambed. No other method could provide an estimate of the thermal properties at this spatial resolution.

Groundwater fluxes were estimated between $2x10^{-6}$ and $4.74x10^{-5}$ $m.s^{-1}$, in very good agreement with the results of the VTPs (Fig. 10). The results suggest a decrease of the groundwater discharge from upstream to downstream, with the most significant inflows located in the first 20-m of the heated section. Elsewhere, groundwater discharge is more diffuse in
space. Results also highlight very punctual increases of groundwater discharge matching with punctual decreases of



streambed temperature SD values, calculated from passive-DTS measurements (Fig. 10). This confirms the possibility of investigating very punctual groundwater inflows and the capability of investigating the spatial evolution of fluxes at very small scale. Thus, active-DTS measurements allow validating the hypothesis made to interpret passive-DTS measurements (diffuse inflows with very punctual increases of groundwater discharge).

The magnitude of groundwater flux is in good agreement with the measured stream flow. Considering the width and the length of the investigated stream where groundwater inflows have been estimated, contribution to the stream can be roughly evaluated around 4 L.s$^{-1}$. This means that 57% of the stream flow at the time of the experiment was contributed by groundwater (the average flow measured at the downstream gauging station was 7 L.s$^{-1}$ for this period).

Results also show (Fig. 10) that the interpretation of active-DTS measurements can be used to improve the
interpretation of passive-DTS measurements. Indeed, the values of thermal conductivities, estimated from active-DTS measurements, can be used to calibrate the inverse model and therefore to reduce the uncertainties on fluxes estimates from passive-DTS measurements. It can lead to a much better quantification of groundwater inflows at a new temporal and spatial scale (although the assumption on the stream temperature remains an issue).

Finally, some limitations and inconveniences should be noted:

(i) The installation of the heated FO cable was entirely manual and rapid, which probably partly explains the large number of measurement points removed before the data interpretation. Nevertheless, the use of tools like ploughs should improve the burying of the cable, limit the alteration of the riverbed and allow for a much better control of the burial depth.

(ii) In gaining streams, the burial depth of the heated cable might potentially affect the thermal response to heating. Indeed, if the cable is too close to the stream, the stream temperature could limit the temperature elevation by dissipating the
heat produced. Further investigations should be done to quantify the effect of the near stream on estimates.

(iii) The data interpretation does not provide the flow direction contrary to the interpretation of passive-DTS measurements. The determination of the flow direction requires coupling the data with complementary approaches, such as passive-DTS measurements, piezometric gradients, etc. The temperature variations recorded before the heating period and after the end of the recovery can also be used as soon as the stream temperature variations are assumed uniform along the
heated section.

(iv) Active-DTS experiments might cause more technical problems than passive-DTS experiments and requires a significant instrumentation (heat-pulse system, electrical cables…). Moreover, the length of the heated section is limited, due to the electrical injection limitation (Simon et al., 2021). Thus, contrary to passive-DTS measurements, the flow investigation at the watershed scale is nearly impossible unless multiplying the installation of heated sections in the
streambed.

(v) Active-DTS measurements provide a punctual estimate of fluxes. To characterize the temporal dynamic of flow, these experiments must be often repeated, which is clearly more constraining than conducting long-term passive-DTS measurements (that require less instrumentation during data acquisition). However, the repetition of active-DTS measurements offer very promising perspectives for environmental monitoring such as recently showed by Abesser et al.



(2020) who repeated surveys under different meteorological or hydrological conditions in order to monitor the thermal and the hydraulic properties of the soil subsurface.

Finally, note also that for the interpretation of both passive- and active-DTS measurements, the flow is assumed to be vertical and perpendicular to the FO cable. Although the absence of nonvertical flow components is generally assumed when using the heat as a tracer (Constantz, 2008; Hatch et al., 2006; Lapham, 1989), nonvertical fluxes can affect natural

temperature profiles and associated fluxes estimates (Bartolino and Niswonger, 1999; Cranswick et al., 2014; Cuthbert and Mackay, 2013; Lautz, 2010; Reeves and Hatch, 2016), and so passive-DTS measurements interpretation. Likewise, nonvertical fluxes could also affect the interpretation of active-DTS measurements. Thus, some studies suggest that the impact of the angle of the flow against the cable is significant as soon as it differs more than ±30° from being perpendicular (Aufleger et al., 2007; Chen et al., 2019; Perzlmaier et al., 2004).

**4.3 Groundwater/stream water exchanges**

The use of these two DTS approaches allow characterizing spatial and temporal patterns of groundwater/stream water exchanges in order to better describe the hydrological behaviour in this headwater watershed. Results and associated fluxes estimates are consistent with previous studies that predicted that 80-90% of the stream flow was induced by groundwater discharge (Fovet et al., 2015b; Martin, 2003).

First, results suggest that the groundwater contribution is concentrated in the wetland and more especially in the upstream near the spring, where the steepest slopes can be observed. Further downstream, beyond 60 m from the spring, the groundwater discharge decreases progressively and rapidly. This confirms the importance of the topography in the stream generation in headwater area (Harvey and Bencala, 1993; Sophocleous, 2002; Tóth, 1963; Winter, 2007) and the role of local topography variations on groundwater discharge (Baxter and Hauer, 2000; Flipo et al., 2014; Frei et al., 2010; Stonedahl et

al., 2010; Tonina and Buffington, 2011; Unland et al., 2013).

Beyond the role of topography which acts as the main driver of groundwater inflows, variations of hydraulic conductivity could also explain the presence of local hotspots with high groundwater inflows, highlighted with both methods in the wetland area, upstream near the spring. Indeed, these hotspots peaks that would highly contribute to the stream flow may be driven by local changes in the hydraulic gradient, induced by the successive streambed topography changes, but are

more likely due to hydraulic properties changes given the amplitude and scale of variations. Such hydraulic conductivity variations could come from uneven bedrock weathering or to the presence of fractures which is very common in such bedrock geology (Buss et al., 2008; Guihéneuf et al., 2014). Such heterogeneities may control flow in the subsurface but can also influence the nature of the streambed. This would also explain why the values of fluxes seem correlated, at least at some places, with the values of thermal conductivities (Fig. 9). Indeed, our results suggest that local hotspots with high

groundwater inflows are also associated to higher values of thermal conductivities. This is consistent with a change of streambed properties. Indeed, clay and silt have much lower hydraulic conductivities than sand but also lower thermal conductivities (Stauffer et al., 2013). Although cross-correlation analysis would be useful to go further in the interpretation,



such correlation would not be surprising since the nature of the streambed affects both its hydraulic conductivity and its thermal properties.

Then, three different time periods were clearly identified with the passive-DTS measurements according to the behaviour of the streambed temperature evolution over time (Fig. 4, 5 and 6). It seems that the increase in precipitations in winter leads to increase gradually the hydraulic gradient, which induces groundwater exfiltration into the stream once the elevation of the groundwater table becomes higher than the elevation of the stream stage. During the spring, the groundwater table decreases progressively and so do the groundwater contribution to stream flow. Since changes in piezometric levels are

periodic with alternating periods of high and low water table levels, we can assume that exchanges have a similar temporal dynamic from year to year, which can help managing water resources.

    These results are consistent with the temporal dynamic of exchanges observed under temperate climate where the intensity of groundwater/surface water exchanges fluctuates according to seasonal patterns (Brunke and Gonser, 1997; Sophocleous, 2002). More important, these results highlight the interest of the long-term monitoring of streambed

temperature with the DTS. The high-frequency of measurements allow monitoring the rapid change of hydrological conditions and therefore to precisely identify the moments when groundwater inflows to the stream. Thus, long-term DTS measurements can provide the precise identification of hotspots of groundwater discharge and of "hot-moments", corresponding to the groundwater discharge periods. These results are useful to understand the hydrological behaviour in the watershed but also for studying biogeochemical hotspots and hot moments (Krause et al., 2017; Singh et al., 2019; Trauth

and Fleckenstein, 2017).

## 5 Conclusions

    Passive- and active-DTS measurements were conducted concurrently in the same experimental site in order to characterize the spatial and temporal patterns of groundwater/stream interactions of a second-order stream. The long-term passive-DTS measurements and the associated quantification of vertical fluxes demonstrated the potential but also the limits

of the approach to characterize both the temporal and spatial variability of groundwater discharge into the stream at the headwater stream scale. Thus, results highlighted preferential discharge areas depending on the streambed properties and topography, showing that the head of the watershed highly contributes to stream flow. The temporal variations of groundwater inflows over time were also clearly identified depending on the annual hydrological cycle and on changes in meteorological and hydrological conditions at the watershed scale. These results are consistent with the results of Martin

(2003) showing that hydraulic gradients on the hillslope of the watershed imply a high contribution of the shallow groundwater to the stream discharge during high water periods.

    However, this study also demonstrates the limitations of passive-DTS measurements. When a single FO cable is buried in the streambed, the data interpretation requires making strong assumptions about the thermal conductivity of sediments and about the stream temperature. Uncertainties may be reduced if previous and independent measurements of the



variability of streambed thermal conductivities are available, through active-DTS measurements for instance. However, the interpretation of passive-DTS measurements would still rely on the assumption that the stream temperature is uniform along the studied section. In practice, the proper estimation of passive-DTS measurements conducted in streambeds would require measuring the stream temperature with the same spatial resolution and thus deploying a second FO cable at the bottom of the stream.

Active-DTS measurements allowed going way further in the characterization of groundwater inflows through the estimate of fluxes and their spatial distribution with a very low uncertainty in comparison with passive-DTS measurements and *in situ* thermal methods. Firstly, the active-DTS experiment allowed identifying and quantifying the high variability of thermal conductivity in space. Secondly, the quantification of groundwater fluxes through the active-DTS measurements clearly showed the co-existence of local hotspots, characterized by very localized and high groundwater inflows and more 670 diffuse groundwater inflows elsewhere along the heated section. The method considerably decreases the uncertainty on fluxes estimates and allows describing the variability of streambed properties at an unprecedented scale.

The complementarity of passive- and active-DTS measurements to fully characterize the dynamic of groundwater/surface water interactions should also be highlighted. Indeed, while long-term passive-DTS measurements can be relatively easily conducted, the application of active-DTS measurements in the field is more constraining, meaning that 675 active-DTS experiments are generally punctual and provide a single punctual estimate of the variability of thermal conductivity and fluxes. Thus, passive-DTS measurements permitted to locate the spatial and temporal patterns of groundwater inflows on relatively large distances, while active-DTS measurements allowed a much more precise and robust estimate of both thermal conductivities and fluxes which can highly contribute to improve passive-DTS methods interpretation.

The combination of active- and passive-DTS methods provided an imaging of the spatial variability of groundwater inflows. It allowed better inferring the role of topography, which acts as the main driver of groundwater inflows in the upper part of the watershed, and also the impact of hydraulic conductivity variations which may explain the presence of very localized and high groundwater inflows. Thus, these methods and especially active-DTS measurements conducted in the streambed open very promising perspectives for novel characterization of the groundwater/stream interfaces, especially if 685 surveys are repeated under different meteorological or hydrological conditions.

**Data availability**

The data presented in this study are available online.

Link through data related to the active-DTS experiment: http://geowww.agrocampus-ouest.fr/geonetwork/apps/georchestra/?uuid=535a3738-0ed7-4376-99f1-9a7a652b893d

Link through data related to the passive-DTS experiment: http://geowww.agrocampus-ouest.fr/geonetwork/apps/georchestra/?uuid=a5f2a68f-bf63-469c-839b-1e1edf1f8624





**Supplement link:**

**Author contribution**

Conceptualization: NS, OB, LL, OF and ZT; Data curation: NS; Formal analysis: NS; Funding acquisition: OB;
Investigation: NS, OB, NL, MF, OF, HLL and ZT; Methodology: NS; Project administration: OB, OF and ZT; Resources:
NL and MF; Software: NS; Supervision: OB; Validation: NS and OB; Visualization: NS; Writing – original draft
preparation: NS; Writing – review & editing: NS, OB, OF, ZT and LL.

**Competing interests**

The authors declare that they have no conflict of interest.

**Acknowledgements**

This research was funded by the Agence de l'Eau Loire Bretagne and by the ANR project EQUIPEX CRITEX (grant number
ANR-11-EQPX-0011).

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
