# Peer review of "Combining passive- and active-DTS measurements to locate and quantify groundwater discharge variability into a headwater stream"

_Hydrology and Earth System Sciences, 2021_

## Referee Comment (RC1)

The analysis of passive heat tracer test is not sufficient. The manuscript describes an application of a passive and an active heat tracer test.

- The manuscript makes a lot of references to other studies of passive heat tracer tests but their findings are not well enough incorporated. Alternatively, present it as a case study.
- It needs to be clear from the start that you only discuss river sections that are gaining water. As rivers are meandering, even rivers that are mainly gaining water can locally lose water. It is stated in the title, but a sentence on it in the abstract would help the reader.
- I would have liked to see a detailed description of the initial boundary and boundary conditions of both the active and passive heat tracer test. And to which extent the simplifications of initial/boundary condition affect the estimated fluxes. A few lines are devoted to this at the end but I see this as something essential that should be part of the beginning.
- In the presented passive heat tracer test, the temperature at the location of the fiber is governed by heat transported by groundwater flowing into the river. No/little attention is paid to heat entering via the river's water surface (and accounting for variations in water depth). It is mentioned that the water levels vary with time (L314).
- Seasonal variations in water depth are not discussed. I can imagine that the section next to the golf court is more canalized then next to the wetlands, and therefore reacts stronger to variations in discharge. Might this also affect the temperature variations and standard deviations presented in Figure 4?
- Image an aquifer in which the water moves as a plug flow (golf court?) of which the water has a narrow distribution of the residence time. This would mean that the seasonal temperature variation of the infiltrating water is found with little damping. In the wetlands I am expecting a much broader distribution of the residence time and much more damping of the temperature of the water that seeps into the river. For a river that gains the same along their length, I would already expect more temperature variation at locations where water moves as a plug flow, with a higher SD (Fig. 4).
- Punctual means exactly on time. In the manuscript it is often used to refer to an exact location.
- The number of measurement locations is too limited to quantify the flow, as is concluded in an early stage of the analysis. As this manuscript is posed as a discussion paper the passive heat tracer test does not seems to contribute much to the discussion.
- The reduction from a complicated river system to a 1D flow model is not well supported with arguments. Apart from all the uncertainties introduced by this simplification, rough estimations of thermal properties introduce additional uncertainty.
- I would like to see the heat tracer test results including the uncertainty in the depth the cable is buried. This is now presented as an afterthought in the discussion. I would like to see it incorporated from the start.

- After accounting for all the uncertainties, are you still confidently able to discuss estimates, or maybe weaken/loosen the goals by discussing locations of inflow. Which already an achievement on its own, considering the difficulty of this type of fieldwork.

[Figure]

Hydrology and
Earth System
Sciences

[Figure]

Discussions

**Combining passive- and active-DTS measurements to locate and quantify groundwater discharge into streams**

Nataline Simon[1], Olivier Bour[1], Mikaël Faucheux[2], Nicolas Lavenant[1], Hugo Le Lay[2], Ophélie Fovet[2], Zahra Thomas[2] and Laurent Longuevergne[1]

[1]Univ Rennes, CNRS, Géosciences Rennes, UMR 6118, 35000 Rennes, France
[2]UMR SAS, INRAE, Institut Agro, Rennes, France

*Correspondence to*: Nataline Simon (nataline.simon2@gmail.com) and Olivier Bour (olivier.bour@univ-rennes1.fr)

**Abstract.** FO-DTS (Fiber Optic Distributed Temperature Sensing) technology has been widely developed to quantify exchanges between groundwater and surface water during the last decade. In this study, we propose, for the first time, to
combine long-term passive-DTS measurements and active-DTS measurements in order to highlight their respective potential to locate and quantify groundwater discharge into streams. On the one hand, passive-DTS measurements consist in monitoring natural temperature fluctuations to detect and localize groundwater inflows and characterize the temporal pattern of exchanges. Although easy to set up, the quantification of flux with this approach often remains difficult since it relies on energy balance models or on the coupling of distributed temperature measurements with additional punctual measurements.
On the other hand, active-DTS methods, recently developed in hydrogeology, consist in continuously monitoring temperature changes induced by a heat source along a FO cable. Recent developments showed that this approach, although more complex to set up than passive-DTS measurements, can address the challenge of quantifying groundwater fluxes and their spatial distribution. Yet it has almost never been conducted in streambed sediments. In this study, both methods are combined by deploying FO cables in the streambed sediments of a first- and second-order stream within a small agricultural
watershed. A numerical model is used to interpret passive-DTS measurements and highlight the temporal and spatial dynamic of groundwater discharge over the annual hydrological cycle. We underline the difficulties and the limitations of deploying a single FO cable to investigate groundwater discharge and show the impact of uncertainty on sediments thermal properties on the quantification of groundwater inflows. On the opposite, the active-DTS experiment allows estimating the spatial distribution of both the thermal conductivity and the groundwater flux at high resolution with very low uncertainties
all along the heated section of FO cable. Our results highlight the added values of conducting active-DTS experiments, eventually combined with passive-DTS measurements, to fully investigate and characterize patterns of groundwater-stream water exchanges at the stream scale. The combination of both methods allows discussing the impact of topography and hydraulic conductivity variations on the variability of groundwater inflows in headwater catchments.
* * *
**Summary of Comments on hess-2021-293.pdf**

**Page: 1**

Author: bfdestombe   Subject: Sticky Note   Date: 27/06/2021, 09:03:00
I would like the abstract to answer the question why passive measurements are not as useful for analysis.

Author: bfdestombe   Subject: Sticky Note   Date: 10/07/2021, 10:06:03
Take the reader by the hand and stress in this abstract that this article only focuses on the gaining sections of a river

Author: bfdestombe   Subject: Sticky Note   Date: 27/06/2021, 08:51:32
Be more specific. estimation from active measurements also relies on energy balance models

Author: bfdestombe   Subject: Sticky Note   Date: 27/06/2021, 08:52:23
point?

Author: bfdestombe   Subject: Sticky Note   Date: 27/06/2021, 08:55:08
Not satisfied with this sentence. Based on your title I would expect this sentence to be (extended over multiple sentences) with more information.

Author: bfdestombe   Subject: Sticky Note   Date: 27/06/2021, 08:59:35
Vague. Be specific.

Author: bfdestombe   Subject: Sticky Note   Date: 27/06/2021, 09:03:50
Use absolute terms

[revised manuscript text omitted]

* * *
Author: bfdestombe   Subject: Sticky Note   Date: 27/06/2021, 09:09:21
Why is there temperature signal lost with large flows?

Author: bfdestombe   Subject: Highlight   Date: 10/07/2021, 12:17:32
Why would that be?

Author: bfdestombe   Subject: Highlight   Date: 10/07/2021, 09:39:23
Please reformulate this sentence

Author: bfdestombe   Subject: Sticky Note   Date: 10/07/2021, 09:46:52
The remainder of this paragraph is also confusingly formulated. The connection between the sentences should be better.

Author: bfdestombe   Subject: Highlight   Date: 10/07/2021, 09:48:21
How do enenrgy balance models differ from passive-DTS measurements

Author: bfdestombe   Subject: Sticky Note   Date: 27/06/2021, 09:11:11
point measurements

Author: bfdestombe   Subject: Sticky Note   Date: 27/06/2021, 09:14:47
Reformulate sentence. And I think you mean to interpolate. Extrapolating is just as difficult with DTS

Author: bfdestombe   Subject: Sticky Note   Date: 27/06/2021, 09:13:23
Merge with previous sentence

Author: bfdestombe   Subject: Sticky Note   Date: 27/06/2021, 09:15:54
Rewrite sentence

Author: bfdestombe   Subject: Sticky Note   Date: 27/06/2021, 09:20:50
With diffusion I tend to agree, but with advection I do not. Be more specific, tell us why and use quantifiable arguments.

Author: bfdestombe   Subject: Highlight   Date: 10/07/2021, 09:50:56
Temperature measured where?

Author: bfdestombe   Subject: Highlight   Date: 10/07/2021, 10:08:10
Merge with previous sentence

[revised manuscript text omitted]

* * *
**Page: 7**

Author: bfdestombe   Subject: Sticky Note   Date: 27/06/2021, 09:45:19
You should be looking for phase shifts, not absolute temperature differences.

Author: bfdestombe   Subject: Sticky Note   Date: 27/06/2021, 09:47:09
What happens to the measurements if you bury the cable opposed to on top of the bed?

Author: bfdestombe   Subject: Sticky Note   Date: 27/06/2021, 09:48:59
Stating here that burying is a possibility requires you to also say also something about placing the cable on top of the bed. If this is not what you intended, please rephrase the sentence.

Author: bfdestombe   Subject: Sticky Note   Date: 27/06/2021, 09:54:10
At locations in the riverbed where the river is gaining, the riverbed is soft and at where the river is losing the riverbed is harder. Burying cables is better achievable in for locations the river is gaining. Be carefull about this, because it introduces a bias in the published measurements.

Author: bfdestombe   Subject: Sticky Note   Date: 27/06/2021, 09:56:07
Can you actually say something about the general applicability of a passive tracer test with half a year of measurements?

Author: bfdestombe   Subject: Highlight   Date: 10/07/2021, 10:17:10
Don't you think that the soil structure of the riverbed coresponds/correlates to the infiltrating behavior. (riverbed of a losing stream is harder than of a gaining stream). This would introduce a bias in your measurements

Author: bfdestombe   Subject: Highlight   Date: 10/07/2021, 10:21:30
Be more precise in your formulation (e.g., the standard error between DTS and RBR)

[revised manuscript text omitted]

---

## Author Comment (AC1)

The reviewer's comments are in *italics* and the responses in regular text.

*I recommend this manuscript to be published after making several major changes. First of all, I would like to congratulate the researchers on their great work. Estimating seepage rates is incredibly hard and this work helps others in their estimation. I recommend several changes that, from my point of view, would help readers/others.*

Dear Bas des Tombe,

First, we would like to thank you very much for your recommendations and the time spent to comment our article. We are pleased to read that you appreciated our work and its implications. Your constructive comments allowed us to clearly identify the points that need to be improved in the manuscript. We will make sure that these points will be addressed and included in the revised version of the manuscript.

Please find in the following responses to your main comments and concerns.

Kind regards,

Nataline Simon & co-authors

First, several comments were about the sources of uncertainties, the uncertainties on estimates, the boundary conditions, the possibility to quantify seepage fluxes given these uncertainties or the meaning and representativity of the estimates. Instead of responding to each comment separately, which would have led to many repetitions, we gathered all these comments below and provide then a general response.

*- The uncertainty of the estimates is discussed at the end (..2 cm offset of the fiber introduces an error in the seepage estimate of 50%..) to a (very) limited extent. It is currently presented as an afterthought and I would like to see this as a significant part of the body of the manuscript. The uncertainty of the estimated seepage rates is an important part of the discussion and allows for better comparison of the different methods. I expect the uncertainty of the estimates to be so large that I wonder if estimating flow is even possible with the presented methods, and it would rather be identifying locations where the river is gaining. Maybe the manuscript is better of estimating locations where the river is gaining, it would be a very valuable contribution to the field.*
*- After accounting for all the uncertainties, are you still confidently able to discuss estimates, or maybe weaken/loosen the goals by discussing locations of inflow. Which already an achievement on its own, considering the difficulty of this type of fieldwork.*
*- I would have liked to see a detailed description of the initial boundary and boundary conditions of both the active and passive heat tracer test. And to which extent the simplifications of initial/boundary condition affect the estimated fluxes. A few lines are devoted to this at the end but I see this as something essential that should be part of the beginning.*
*- The reduction from a complicated river system to a 1D flow model is not well supported with arguments. Apart from all the uncertainties introduced by this simplification, rough estimations of thermal properties introduce additional uncertainty.*
*- I would like to see the heat tracer test results including the uncertainty in the depth the cable is buried. This is now presented as an afterthought in the discussion. I would like to see it incorporated from the start.*

The manuscript describes and compares the application of active and passive heat tracer tests to locate and/or estimate groundwater discharge into streams. Here, you are perfectly right that the question of the uncertainties of the estimates must be addressed. To achieve this, it is important to clearly separate both approaches (passive vs. active).

Concerning ***the passive experiment***, we fully agree with your comment. In the manuscript, we conclude that the analysis of the passive heat tracer test is not sufficient to estimate fluxes since the uncertainties of the estimates are too large with this approach. Thus, we explain that this approach is more suitable for identifying locations where the river is gaining. As you mentioned in your comments, the uncertainties of the estimates for passive experiments are due to several sources of uncertainties, especially:
1) the uncertainties on the thermal properties of the sediments, on which we mainly focused in the manuscript. We widely discuss the projection of the uncertainty of the thermal conductivity on the estimate of the seepage rates and clearly show (for instance in the section 3.1.3 and in figure 9) that the assumption on thermal conductivities highly affects the fluxes estimates.
2) the uncertainty on the burial depth of the cable, which is rapidly addressed in the manuscript as you correctly underlined.
3) the uncertainties linked to the boundary conditions of the model, which are not directly discussed in the manuscript.

Because the uncertainty of estimates resulting from the uncertainty on thermal conductivities is so large, we considered and concluded that the passive experiment is not sufficient to estimate seepage rates. Considering this, it appears that discussing more about the assumptions on the burial depth of the cable and the boundary conditions would be not actually relevant (the quality of the estimates being not satisfactory enough). Of course, it would have been very interesting but we think that such sensitivity analysis would overload the manuscript, which is in itself rather long, and would digress from the main objective of the study, which is the comparison of active and passive experiments.

However, we are keen to strengthen these points in the revised version of the manuscript. Thus, the sources of the uncertainties would be clearly identified and each point would be discussed.

Concerning the ***active experiment*** however, we are convinced that the approach allows locating **and estimating** fluxes. Indeed, compared to the passive experiment, the uncertainties are very low and the estimates are significant. Here are the reasons why:

First, as explained in the manuscript, the analysis of the temperature evolution during heating periods leads to estimate as a first step the thermal conductivity of the sediments with a very good accuracy (as demonstrated in Simon et al., WRR 2021). Thus, the approach does not require assuming the value of the thermal conductivities which considerably reduce the uncertainties of the fluxes estimates. Note also that in our previous study (Simon et al., WRR 2021), the sensitivity of different parameters (thermal properties, groundwater flow, heat injection) on the temperature rise has already been studied in details. Errors induced by noise in temperature measurements were also discussed in this study.

Then, concerning the burial depth of the cable, we mentioned in the manuscript that it might potentially affect the thermal response to heating, if the cable is too close to the stream and we explained that further investigations should be done to quantify the effect of the near stream on estimates. Here, we are confident that the burial depth has low effects on the temperature evolution. The active experiment was conducted straight after the installation of the cable, ensuring that the burial depth was sufficient to limit the effect of the near stream (results from modeling showed that the heating is particularly localized around the heated cable).

Then, one of the main advantages of the application of the active experiment in gaining streams is that the temperature evolution is independent of temperature boundary conditions, as long as the groundwater temperature is constant over time (which is the case here). Indeed, in gaining conditions, the stream temperature variations do not (or just a few) propagate in depth meaning that the temperature variations recording along the FO cable would exclusively due to the heat injection. However, it is essential to confirm this assumption by

analyzing the temperature variations recorded in non-heated buried sections of the FO cable. This verification was made but is not clearly explained nor described in the manuscript. This will be corrected in the revised manuscript. In losing conditions, since diurnal water temperature variations propagates deeper, it would be necessary to separate the temperature evolution induced by the heat injection and the one depending on stream temperature variations.

Except for the thermal conductivity, it is true that the potential sources of uncertainties are not clearly identified in the manuscript. This will be improved in the revised manuscript and we will make sure that the effects of both the burial depth of the cable and the temperature boundary conditions will be discussed in order to strengthen the conclusion about the efficiency of active experiment for estimating seepage fluxes.

*Thus, how to go from a complex 3D world to a highly simplified 1D model.*

The use of simplified 1D model is classic when using heat as a tracer. For passive experiments, this was already proposed in many applications (Anderson, 2005; Constantz, 2008; Goto et al., 2005; Hatch et al., 2006; Keery et al., 2007; among others). The FLUX-BOT model used to interpret passive-DTS measurements was proposed by Munz and Schmidt (2017) especially for quantifying water fluxes between groundwater and surface water. The main assumption of the model is that fluxes are strictly vertical to the stream, which is a classical assumption made in all studies using heat to quantify seepage fluxes. We are aware this is a main issue but this is why this point is discussed in the discussion section. The same assumption is made when interpreting active-DTS measurements. Thus, it is true that it is not possible yet to estimate groundwater seepages the 3D, but it allows at least estimating fluxes in the direction perpendicular to the river, which should be the main direction of fluxes. However, it should be noted that the measurements achieved during active experiments are still significant and can therefore be interpreted.

*The manuscript makes a lot of references to other studies of passive heat tracer tests but their findings are not well enough incorporated. Alternatively, present it as a case study.*

You are perfectly right. First, the interpretation of passive-DTS measurements allows discussing the effect of the assumption on thermal conductivities on estimates, which is seldom done in studies involving passive heat tracer tests. Then, we show how the results of the active-DTS experiment, and especially the estimate of the thermal conductivity, can be used to improve the fluxes estimates from passive measurements. Thus, the introduction will be improved in order to highlight these points and the originality of our work compared to other passive heat tracer tests.

*It needs to be clear from the start that you only discuss river sections that are gaining water. As rivers are meandering, even rivers that are mainly gaining water can locally lose water. It is stated in the title, but a sentence on it in the abstract would help the reader.*

You are right. This point is essential since the effects of both the burial depth of the cable and the temperature boundary conditions are different in losing and gaining conditions. This will be included in the revised version of the abstract. The application of the approach would also be more clearly discussed in the discussion section.

*In the presented passive heat tracer test, the temperature at the location of the fiber is governed by heat transported by groundwater flowing into the river. No/little attention is paid to heat entering via the river's water surface (and accounting for variations in water depth). It is mentioned that the water levels vary with time (L314).*

Concerning stream temperature variations, this is important to distinguish the passive and the active experiments.

Stream temperature variations are fully considered when interpreting the passive-DTS measurements, since the stream temperature measured at the bottom of the stream is used as the upper boundary condition of the model. The model allows deducing groundwater fluxes into the stream, and therefore gradient variations, from temperature changes.

For active-DTS measurements conducted in gaining conditions, we assume that the temperature evolution is independent of the stream temperature variations that do not (or just a few) propagate in depth. However, you are right that in losing conditions, since diurnal water temperature variations propagates deeper, stream temperature variations could affect the measured temperature evolution. In this case, it could be necessary to separate the temperature evolution induced by the heat injection and the one depending on stream temperature variations. This point is not clearly explained in the manuscript and will be added in the discussion of the revised manuscript.

Either in gaining or losing conditions, we suggest monitoring the "natural" temperature variations along a buried section of the FO cable. If temperature variations are recorded in non-heated sections during heating periods, the user could filter the data accordingly (especially in losing conditions). This was made in this study, as explained in the supplement material (section 3.1) :

"During this period, temperature has been also recorded in sediments with the non-heated FO cable and shows an average temperature of 12.1 °C and a standard deviation of 0.12 °C. This shows that i) the streambed temperature is not affected by potentials air/stream variations during the experiment duration, meaning that the temporal variations are exclusively due to the heat experiment and ii) the heat experiment induces only a small and local thermal perturbation of the streambed around the buried FO cable."

This point will be more clearly explained in the revised manuscript.

*Seasonal variations in water depth are not discussed. I can imagine that the section next to the golf court is more canalized then next to the wetlands, and therefore reacts stronger to variations in discharge. Might this also affect the temperature variations and standard deviations presented in Figure 4? Image an aquifer in which the water moves as a plug flow (golf court?) of which the water has a narrow distribution of the residence time. This would mean that the seasonal temperature variation of the infiltrating water is found with little damping. In the wetlands I am expecting a much broader distribution of the residence time and much more damping of the temperature of the water that seeps into the river. For a river that gains the same along their length, I would already expect more temperature variation at locations where water moves as a plug flow, with a higher SD (Fig. 4)*

Indeed, the assumption of broader residence times in the wetland could potentially explain the evolution of the SD. However, similar stream temperature variations are observed in the wetland and in the golf area. It would require very short residence times in the golf area to obtain such results - the residence times in the wetland being high as shown by the constant temperature signal measured in the piezometer near the stream in the wetland. Thus, it does not seem very consistent since it would imply very peculiar behavior.

This assumption could be proved or disproved by modeling the watershed considering a broader distribution of residence times in the wetland. However, such model would be difficult to achieve and time-consuming. Moreover, we are not sure that it would be fully relevant, since previous studies showed that the behavior of the aquifer is relatively "simple" and did not highlight that the water moves as a plug flow in the golf court.

*Punctual means exactly on time. In the manuscript it is often used to refer to an exact location.*

Sorry about this confusion. This will be corrected in the revised manuscript.

*The number of measurement locations is too limited to quantify the flow, as is concluded in an early stage of the analysis. As this manuscript is posed as a discussion paper the passive heat tracer test does not seems to contribute much to the discussion*

Sorry, but we don't really understand this comment. For passive-DTS measurements, the FLUX-BOT model was applied on almost all data collected along the cable deployed in streambed sediments in the wetland area, for d < 150 m, which represents 545 measurements points. Thus, although the model is not applicable for d > 150 m, the results of the passive heat tracer test remain relevant for the discussion. Concerning active-DTS measurements, fluxes were estimated for 172 measurements points along a 60-meter section of cable, which is also an excellent performance. To our knowledge the first time that active-DTS measurements have been applied along such a section. To better highlight the number of measurements relevant to each method, we propose to add this information in the revised version of supplementary materials.

Once again, we thank you very for your comments and the time spent to review our article. We will make sure that all your concerns and all the comments included in the .pdf file will be addressed and included in the revised version of the manuscript.

---

## Author Response (AR1)

The reviewer's comments are in *italics* and the responses in regular text.

*Editor decision: Reconsider after major revisions (further review by editor and referees) by Miriam Coenders-Gerrits*

*As can be seen the two reviewers appreciate your study. Nonetheless, they have quite some comments which should be addressed. Especially, they comment on the quantification of the fluxes, which is currently under-emphasised. Please consider this, as well as a possible refocus of the manuscript. Furthermore, they recommend to elaborate on the uncertainties. Related to this in de case of active DTS: depending on the flow rates and the chosen heating rate of the active DTS-cable the measuring error is different. See for example the work of Van Ramshorst et al (https://doi.org/10.5194/amt-13-5423-2020). No need to cite this paper (sorry self-referencing), but maybe use it as a reference for yourself for further studies on this topic.*

Dear Miriam Coenders-Gerrits,

Thank you for your reply and for considering our study as a potential publication for HESS. We were very pleased to read the enthusiasm of both reviewers. We are very grateful for the time they spend to review our manuscript. We found their comments particularly constructive and relevant.

In the first version of the manuscript, we tried to carefully provide uncertainties of estimates as for the interpretation of passive-DTS measurements as for the interpretation of the innovative active-DTS experiment. In fact, the main advantage of active-DTS experiments is to provide estimates with low uncertainties for a large range of groundwater discharge values (Simon et al., 2021). Note that such low uncertainties are possible because the physical processes controlling flow and heat transport in porous media (fully described in Simon et al., 2021) are different than the one controlling heat transport in fluids, for instance in water flowing through open boreholes (Read et al., 2014) or in air while measuring wind speed (Van Ramshorst et al., 2020). However, the recommendations of the reviewer clearly show that some clarifications and a reorganization of our manuscript were required to better highlight our main contribution and to avoid any misunderstanding. Thus, following their suggestions, we reorganize and made significant revisions of our manuscript to clarify the objectives and discuss the interest and limitations of both approaches for estimating groundwater discharges in light of uncertainty estimates.

Our first goal was to refocus the manuscript on the main research question that is the quantification of groundwater discharge. To do so, we fully rewrote the abstract and a large part of the introduction. Then, we considerably reduced the result section dealing with the qualitative interpretation of passive-DTS measurements. Lastly, we fully rewrote and rearranged the discussion section. In the initial version of the manuscript, the discussion was mostly about the methodology and the limits of each approach. In the revised version, we focused on research questions and showed how each approach could be used to describe hydrological processes.

Secondly, the question of uncertainties on fluxes estimates is now fully discussed both in the results and discussion sections. In the results section, uncertainty on estimates are provided for each estimation of groundwater discharge and discussed in the text. To make it clear, we also detail in the discussion section the sources of uncertainties for each approach and tried to assess their effects on fluxes estimates.

Please find below a detailed point-by-point response to reviewers' comments. As you may see, we carefully answer to every single comment/question addressed. We sincerely think all these revisions (as major as minor revisions) contributed to improve our study and are grateful to the reviewers for these improvements.

Thank you for your time,

Kind regards,

Nataline Simon & co-authors.

*Response to the comments of Bas des Tombe (Reviewer 1)*

*I recommend this manuscript to be published after making several major changes. First of all, I would like to congratulate the researchers on their great work. Estimating seepage rates is incredibly hard and this work helps others in their estimation. I recommend several changes that, from my point of view, would help readers/others.*

Dear Bas des Tombe,

First, we would like to thank you very much for your recommendations and the time spent to comment our article. We are pleased to read that you appreciated our work and its implications. Your constructive comments allowed us to clearly identify the points that need to be improved in the manuscript. We carefully addressed all these points in the revised version of the manuscript.

Please find in the following, a point-by-point response to your comments. Concerning major concerns, a detailed response has ever been provided during the interactive public discussion. Thus, we focus here on the presentation of the revisions made to response to comments.

Kind regards,

Nataline Simon & co-authors
* * *
*Responses to main comments*

Several comments from the reviewers were about the sources of uncertainties, the uncertainties on estimates, the boundary conditions, the possibility to quantify seepage fluxes given these uncertainties or the meaning and representativity of the estimates. Instead of responding to each comment separately, which would have led to many repetitions, we gathered all these comments below and provide a general response.

➤ *The uncertainty of the estimates is discussed at the end (..2 cm offset of the fiber introduces an error in the seepage estimate of 50%..) to a (very) limited extent. It is currently presented as an afterthought and I would like to see this as a significant part of the body of the manuscript. The uncertainty of the estimated seepage rates is an important part of the discussion and allows for better comparison of the different methods. I expect the uncertainty of the estimates to be so large that I wonder if estimating flow is even possible with the presented methods, and it would rather be identifying locations where the river is gaining. Maybe the manuscript is better of estimating locations where the river is gaining, it would be a very valuable contribution to the field.*
*- After accounting for all the uncertainties, are you still confidently able to discuss estimates, or maybe weaken/loosen the goals by discussing locations of inflow. Which already an achievement on its own, considering the difficulty of this type of fieldwork.*
*- I would have liked to see a detailed description of the initial boundary and boundary conditions of both the active and passive heat tracer test. And to which extent the simplifications of initial/boundary*

*condition affect the estimated fluxes. A few lines are devoted to this at the end but I see this as something essential that should be part of the beginning.*
*- The reduction from a complicated river system to a 1D flow model is not well supported with arguments. Apart from all the uncertainties introduced by this simplification, rough estimations of thermal properties introduce additional uncertainty.*
*- I would like to see the heat tracer test results including the uncertainty in the depth the cable is buried. This is now presented as an afterthought in the discussion. I would like to see it incorporated from the start.*

The manuscript describes and compares the application of both active and passive heat tracer tests to locate and/or estimate groundwater discharge into streams. Here, you are perfectly right that the question of the uncertainties of the estimates should be more carefully addressed. To achieve this, we revised the manuscript in order to clearly identify and discuss more specifically, for each approach, the source of uncertainties. These changes allow discussing the relevance of fluxes estimates and strengthen the interest of developing active-DTS measurements in such context.

Thus, now, uncertainties on estimates are presented more clearly when presenting the results about groundwater discharges. Moreover, the discussion section was fully rewritten and rearranged in order to address the question of uncertainties on fluxes estimates.

First, concerning passive-DTS measurements, you will find in the revised manuscript (From L. 535 to 577) a full paragraph discussing this point bringing to the conclusion that the uncertainty of estimates resulting from the uncertainty on thermal conductivities is so large that passive-DTS measurements are not sufficient to estimate seepage rates:

[revised manuscript text omitted]

Then, we strengthened the discussion about uncertainties on fluxes estimates while interpreting active-DTS measurements in order to fully demonstrate that the uncertainties are very low and the fluxes estimates significant. Thus, in the discussion section of the revised manuscript, we added the following (From L. 607 to 625):

"It clearly appears that the main advantages of the approach are i) the low uncertainty on fluxes estimates and ii) the associated estimates of thermal conductivities. The use of the ADTS Toolbox, that automatically interprets active-DTS measurements (Simon and Bour, Submitted), highly facilitates data interpretation, which is finally much easier than the interpretation of passive-DTS measurements. Contrary to passive-DTS measurements, the interpretation of active-DTS measurements provides estimates of thermal conductivities with a very good accuracy. Thus, the approach does not require assuming thermal conductivity values which considerably reduce the uncertainties of the fluxes estimates. However, the burial depth of the heated cable might potentially affect the thermal response, if the cable is too close to the stream. In this case, the stream temperature could limit the temperature elevation by dissipating the heat produced and further investigations should be done to quantify the effect of the near stream on estimates. However here, the active-DTS experiment was conducted straight after the installation of the cable, ensuring that the burial depth was sufficient to limit the effect of the near stream (results from modelling showed that the heating is particularly localized around the heated cable). Finally, in gaining streams, active-DTS measurements are independent of temperature boundary conditions, as long as the groundwater temperature is constant over time. Indeed, in gaining

conditions, with no groundwater temperature variations, the temperature evolution measured along the FO cable is exclusively due to heat injection, streambed sediment properties and groundwater flow intensity. Here, over the heating experiment, an average temperature of 12.1 °C with a standard deviation of 0.12 °C has been recorded in sediments along non-heated buried sections of the cable demonstrating that the streambed temperature was not affected by potentials air/stream variations and that the temperature variations recorded along the heated section are exclusively due to the heat experiment. In losing conditions, since diurnal water temperature variations propagates deeper, it could be necessary to separate the temperature evolution induced by the heat injection and the one depending on stream temperature variations."

> *Thus, how to go from a complex 3D world to a highly simplified 1D model.*

The use of simplified 1D model is classic when using heat as a tracer. For passive experiments, this was already proposed in many applications (Anderson, 2005; Constantz, 2008; Goto et al., 2005; Hatch et al., 2006; Keery et al., 2007; among others). The FLUX-BOT model used to interpret passive-DTS measurements was proposed by Munz and Schmidt (2017) especially for quantifying water fluxes between groundwater and surface water. The main assumption of the model is that fluxes are strictly vertical to the stream, which is a classical assumption made in all studies using heat to quantify seepage fluxes. The same assumption is made when interpreting active-DTS measurements. Thus, it is true that it is not possible yet to estimate groundwater seepages the 3D, but it allows at least estimating fluxes in the direction perpendicular to the river, which should be the main direction of fluxes. However, it should be noted that the measurements achieved during active experiments are still significant and can therefore be interpreted.

We are aware this is a main issue but this point is now clearly is discussed in the revised manuscript:

"Moreover, note also that for the interpretation of both passive- and active-DTS measurements, the flow is assumed to be vertical and perpendicular to the FO cable. Although flow is generally assumed vertical when using heat as a tracer for studying groundwater/stream interactions (Constantz, 2008; Hatch et al., 2006; Lapham, 1989), nonvertical fluxes can affect natural temperature profiles and associated fluxes estimates (Bartolino and Niswonger, 1999; Cranswick et al., 2014; Cuthbert and Mackay, 2013; Lautz, 2010; Reeves and Hatch, 2016). Obviously, like for most thermal-based methods, this is a main limitation when using passive-DTS measurements to detect and quantify groundwater discharge. Likewise, nonvertical fluxes could also affect the interpretation of active-DTS measurements. Thus, some studies suggest that the impact of the angle of the flow against the cable is significant as soon as it differs more than ±30° from being perpendicular (Aufleger et al., 2007; Chen et al., 2019; Perzlmaier et al., 2004). " (From L. 641 to 649).

> *The manuscript makes a lot of references to other studies of passive heat tracer tests but their findings are not well enough incorporated. Alternatively, present it as a case study.*

Following this comment, we decided to significantly revise the introduction and especially the last paragraph. This change allows explaining better our contribution compared to previous studies. We hope this change helps to understand why this study is not presented as a case study (From L. 99 to 111):

"In this study, we propose to use for the first time active-DTS measurements to quantify groundwater discharge in the stream of a headwater catchment. The application of active-DTS methods in such

context is particularly promising since the interpretation of active-DTS measurements in saturated porous media provides estimates of both sediments thermal conductivities and groundwater fluxes over a large range and with an excellent accuracy (Simon et al., 2021)., This method should allow quantifying groundwater discharge and characterizing the streambed thermal properties at an unprecedent spatial scale. In complement of active-DTS measurements, which were limited in space and time, a long-term passive-DTS experiment was conducted at the catchment scale in order to infer the location and dynamics of groundwater discharge. Therefore, this study also investigates how these two experiments could be compared and combined to characterize both the spatial and the temporal dynamics of groundwater discharge. To do so, FO cables were deployed in the streambed sediments of a headwater stream within a small agricultural watershed. In the following, we first present the headwater watershed and the experimental setup before presenting the methods used to interpret both passive- and active-DTS measurements. Fluxes estimates obtained with both passive- and active-DTS measurements are then compared and the advantages and limitations of each method are finally discussed"

➢ *It needs to be clear from the start that you only discuss river sections that are gaining water. As rivers are meandering, even rivers that are mainly gaining water can locally lose water. It is stated in the title, but a sentence on it in the abstract would help the reader.*

You are right. This point is essential and was not clearly explained in the initial version of the study. As you may see in the revised version of the manuscript, **we fully rewrote the abstract**. The aim was to refocus the text on the research question that is the fluxes quantification and to propose a more compelling abstract. **While rewriting the abstract, we made sure that it was clear from the start that we focus here on groundwater discharge into streams**. We also propose to **modify the title of the paper to restrengthen this point.**

➢ *In the presented passive heat tracer test, the temperature at the location of the fiber is governed by heat transported by groundwater flowing into the river. No/little attention is paid to heat entering via the river's water surface (and accounting for variations in water depth). It is mentioned that the water levels vary with time (L314).*

Stream temperature variations are fully considered when interpreting the passive-DTS measurements, since the stream temperature measured is used as the upper boundary condition of the model. The model allows deducing groundwater fluxes into the stream, and therefore gradient variations, from temperature changes.

For active-DTS measurements conducted in gaining conditions, we assume that the temperature evolution is independent of the stream temperature variations that do not (or just a few) propagate in depth. This assumption was verified through the analysis of temperature variations recorded along non-heated buried sections of FO cable. However, in losing conditions, since diurnal water temperature variations propagate deeper, stream temperature variations could affect the measured temperature evolution. In this case, it could be necessary to separate the temperature evolution induced by the heat injection and the one depending on stream temperature variations.

These points were not clearly explained in the initial manuscript and are now fully discussed in the revised manuscript (From L. 617 to 625):

"Finally, in gaining streams, active-DTS measurements are independent of temperature boundary conditions, as long as the groundwater temperature is constant over time. Indeed, in gaining conditions, with no groundwater temperature variations, the temperature evolution measured along the FO cable is exclusively due to heat injection, streambed sediment properties and groundwater flow intensity. Here, over the heating experiment, an average temperature of 12.1 °C with a standard deviation of 0.12 °C has been recorded in sediments along non-heated buried sections of the cable demonstrating that the streambed temperature was not affected by potentials air/stream variations and that the temperature variations recorded along the heated section are exclusively due to the heat experiment. In losing conditions, since diurnal water temperature variations propagates deeper, it could be necessary to separate the temperature evolution induced by the heat injection and the one depending on stream temperature variations."

> *Seasonal variations in water depth are not discussed. I can imagine that the section next to the golf court is more canalized then next to the wetlands, and therefore reacts stronger to variations in discharge. Might this also affect the temperature variations and standard deviations presented in Figure 4? Image an aquifer in which the water moves as a plug flow (golf court?) of which the water has a narrow distribution of the residence time. This would mean that the seasonal temperature variation of the infiltrating water is found with little damping. In the wetlands I am expecting a much broader distribution of the residence time and much more damping of the temperature of the water that seeps into the river. For a river that gains the same along their length, I would already expect more temperature variation at locations where water moves as a plug flow, with a higher SD (Fig. 4)*

Indeed, the assumption of broader residence times in the wetland could potentially explain the evolution of the SD. However, similar stream temperature variations are observed in the wetland and in the golf area. It would require very short residence times in the golf area to obtain such results - the residence times in the wetland being high as shown by the constant temperature signal measured in the piezometer near the stream in the wetland. Thus, it does not seem very consistent since it would imply very peculiar behavior.

This assumption could be proved or disproved by modeling the watershed considering a broader distribution of residence times in the wetland. However, such model would be difficult to achieve and time-consuming. Moreover, we are not sure that it would be fully relevant, since previous studies showed that the behavior of the aquifer is relatively "simple" and did not highlight that the water moves as a plug flow in the golf court.

> *Punctual means exactly on time. In the manuscript it is often used to refer to an exact location.*

Sorry for this confusion. This is now changed in the revised manuscript (corrections were adapted according to each context).

> *The number of measurement locations is too limited to quantify the flow, as is concluded in an early stage of the analysis. As this manuscript is posed as a discussion paper the passive heat tracer test does not seems to contribute much to the discussion*

Sorry, but we don't really understand this comment. For passive-DTS measurements, the FLUX-BOT model was applied on almost all data collected along the cable deployed in streambed sediments in the wetland area, for d < 150 m, which represents 545 measurements points. Thus, the results of the passive heat tracer test remain fully relevant for the discussion. Concerning active-DTS measurements,

fluxes were estimated for 172 measurements points along a 60-meter section of cable, which is also an excellent performance.

To be sure everything is clear, the following were added in the revised manuscript in the discussion section:

- For passive-DTS measurements, we now precise: "The inverse numerical FLUX-BOT model was used to quantify the vertical fluxes from passive-DTS measurements in the first 150 meters of the temperature profile which represents 545 measurements points" (From L. 535 to 536)

- For active-DTS measurements:
"Finally, among all the section used for active-DTS measurements, 172 measurements points are thought to be significant." (From L. 271 to 272)
"Despite some difficulties to install the heated FO cable, fluxes were even so estimated for 172 measurements points along a 60-meter section of cable, which is an excellent performance. This high resolution is particularly interesting to characterize spatial variabilities at small scale. Such results cannot be achieved with any other methods commonly used in this context." (From L. 582 to 585)

**Responses to comments attached in the file**

• *Abstract*

➢ *I would like the abstract to answer the question why passive measurements are not as useful for analysis.*

We understand this comment but we don't totally agree. In fact, we don't want to focus the abstract on passive-DTS measurements. Otherwise, it would give the impression that the main topic of the study. However, based on our results, we clearly explain in the abstract the difficulty of quantifying GW discharge (as explained below, we revised the part of the abstract and introduction concerning fluxes quantification through passive-DTS measurements).

*Please note that the rest of comments relative to the abstract in the attached file concern specific sentences that were removed in the revised version of the abstract.*

• *Introduction*

➢ *L. 54 Why is this the case?*
To be more specific, we now precise in the revised manuscript : "Considering the spatial variability and the complexity of flow at the GW/stream interface, extensive information on spatial and temporal temperature patterns are required to gain a more complete understanding of flows at reach scale, and even more at watershed scale." (L. 58 to 60 in the revised manuscript)

➢ *L. 63 Why is there temperature signal lost with large flows ?*
*L. 65 « improving » - Why would that be?*

In large river flows, the GW discharge is very small compared to the stream flow. Thus, GW discharge doesn't necessarily induce temperature anomalies in the stream (the GW temperature signal is "lost" in the stream). This sentence was improved in the revised manuscript in order to clarify its meaning:

> "However their use remains limited because it requires monitoring significant temperature changes over time, limiting the application of the method to large groundwater inflows or small headwater streams. To overcome such limitations, some studies proposed to detect thermal anomalies in the streambed by burying the FO cable into the streambed sediments in the hyporheic zone (Krause et al., 2012; Le Lay et al., 2019b; Lowry et al., 2007), to improve the possibility of localizing groundwater inflows." (L. 70 to 74 in the revised manuscript)

➢ *L. 66 « However, beyond the localization of inflows, the distributed quantification of fluxes from passive-DTS measurements remains even more challenging » - Please reformulate this sentence. The remainder of this paragraph is also confusingly formulated. The connection between the sentences should be better*
*L. 68 « energy balance models » - How do enenrgy balance models differ from passive-DTS measurements*
*L. 73 « which raises the question of the extrapolation » - Reformulate sentence. And I think you mean to interpolate. Extrapolating is just as difficult with DTS*
*L. 73 « It also raises the question of the estimation of associated uncertainties. » - Merge with previous sentence*

Seeing these comments, the paragraph was fully rewritten and reorganized. We hope the revised paragraph is clearer (From L. 69 to 85):

> "Energy balance models have been efficiently applied to interpret passive-DTS measurements and quantify groundwater/stream water exchanges (Selker et al., 2006b; Westhoff et al., 2011). However their use remains limited because it requires monitoring significant temperature changes over time, limiting the application of the method to large groundwater inflows or small headwater streams. To overcome such limitations, some studies proposed to detect thermal anomalies in the streambed by burying the FO cable into the streambed sediments in the hyporheic zone (Krause et al., 2012; Le Lay et al., 2019b; Lowry et al., 2007), to improve the possibility of localizing groundwater inflows.
>
> However, the quantification of fluxes from passive-DTS measurements remains challenging. In theory, the implementation of three FO cables buried at different depths, as proposed by Mamer and Lowry (2013), would be ideal to measure the attenuation of the stream temperature variations into the sediments in order to get high-resolution fluxes estimates. Unfortunately, such approach is technically very difficult to apply in the field. Considering this difficulty, Le Lay et al. (2019a) proposed coupling FO-DTS data collected along a single fiber-optic cable at a given depth with punctual temperature measurements from thermal lances. In the approach, it is assumed that temperature boundary conditions can be characterized from the temperature measurements collected with the thermal lances and extrapolated all along the stream to be combined with FO-DTS measured at a given depth. Moreover, the question of sediments thermal properties and their spatial variability remained unexplored, even though thermal conductivity highly impacts flux estimates (Sebok and Müller, 2019).Based on these assumptions, they showed the temporal variability of exchanges associated to the annual hydrological cycle and the possibility of estimating diffuse groundwater inflows (Le Lay et al., 2019a)."

➤ *L. 76 « Contrary to passive-DTS methods, active-DTS methods, consisting in injecting a heat source in the sediments, better constrain the signature of diffusion and advection along a FO cable (Bense et al., 2016). » - Rewrite sentence*
*L. 77 With diffusion I tend to agree, but with advection I do not. Be more specific, tell us why and use quantifiable arguments.*

This part of this paragraph was rewritten. We decided to remove the part concerning advection and diffusion which was not essential in the introduction (knowing that these processes are fully described below).  In the revised manuscript, you will find this revised paragraph (From L. 86 to 90):

> "Alternatively, active-DTS methods, consisting in heating the FO cable, have been recently developed to improve the capabilities of FO-DTS methods for estimating fluxes in different environmental conditions (Bense et al., 2016; Simon et al., 2021). In particular, it was demonstrated that the difference of temperature between an electrically heated and a non-heated FO cable is directly dependent on water fluxes, offering the possibility to estimate fluxes (Bense et al., 2016; Read et al., 2014; Sayde et al., 2015)"

➤ *L. 78 « the difference of temperature measured » - Temperature measured where?*
The term measured was removed.

• **2.1 The experimental setup on the Kerrien watershed**

➤ *L. 108 « 14% » - What does this percentage mean?*
This corresponds to the slopes, as now specified in the manuscript:

> "The Kerrien sub-watershed is a small agricultural watershed (9.5 ha) with steeper slopes in the upper parts (14% slopes) than in the bottom lands (5% slopes), characterized by a large wetland  (Ruiz et al., 2002)." (From L. 122 to 124)

➤ *L. 123 « negligible » - compared to ?*
The term was replaced by: "Runoff is **insignificant**".

➤ *L. 130 - Be more specific. The head gradient is year round directed towards the river, and thus would expect a year rond gaining river. F1b seems to be located lower. Are you sure that the river is the main drain of this subcatchment? Are F2 and F1b reversed?*
*Fig 2 - F2 and F1b mixed up?*

Seeing these comments, it appears that there is a misunderstanding here, which is probably due to a confusion in the legend of the Figure Fig 2a. When the legend indicates "F2 (2m)" and "F1b (5m)", it doesn't mention the distance between the borehole and the stream but the depth of each borehole. As you can see in Fig 1, F2 is near the stream and F1b higher in the catchment. To avoid this confusion, the legend was modified and now specifies "F2 (2m-depth)", "F1b (5m-depth)" etc:

—F5b (20 m-depth)    — F4 (15 m-depth)    —F2 (2 m-depth)    —F1b (5 m-depth)

➤ *L. 138 « Considering the relatively small difference of temperature between groundwater and stream water, burying the FO cable within the sediments should facilitate the detection of potential*

*temperature anomalies as marker of groundwater discharge » - You should be looking for phase shifts, not absolute temperature differences*
*What happens to the measurements if you bury the cable opposed to on top of the bed? Stating here that burying is a possibility requires you to also say also something about placing the cable on top of the bed. If this is not what you intended, please rephrase the sentence.*

To clarify things, we decided to modify the associated sentences (L. 154 to 160):

"The detection in the stream of thermal anomalies induced by groundwater discharge requires a significant contrast of temperature between stream and groundwater as well as a significant groundwater discharge compared to the stream flow. Here, considering the relatively small difference of temperature between groundwater and stream water and in order to detect potential diffusive inflows, the choice was made to bury the FO cable within the sediments, which should facilitate the detection of potential temperature anomalies as marker of groundwater discharge (Krause et al., 2012; Le Lay et al., 2019b; Lowry et al., 2007). Otherwise, active-DTS measurements should highlight advective heat transfer, controlled by groundwater discharge."

• *Passive DTS measurements and data interpretation*

➢ *L.145 - Can you actually say something about the general applicability of a passive tracer test with half a year of measurements?*

It is true that a longer measurement period would necessary have been interesting. However, having half a year of measurements with such spatial and temperature resolutions (data are collected at 25 cm and 10 minutes sampling interval) is already very remarkable and rare. To our knowledge, only Le Lay et al. (2019) proposed a long-term monitoring of passive-DTS measurements (over one year) in such context. Moreover, our results clearly show the temporal dynamic of exchanges and allow delimiting "hot-moments" corresponding to GW discharge periods, which a very interesting result. To better highlight this point, we improved the paragraph in the discussion section concerning the temporal dynamic of exchanges (From L. 689 to 691):

"Here, with only eight months of measurements, the experiment allowed continuously monitoring hydrological conditions changes and clearly identifying "hot-moments" corresponding to groundwater discharge periods**."**

In complement, the following was added in the perspectives (From L. 728 to 732):

"Being able to continuously monitoring the temporal dynamic of exchanges is a very promising achievement that could be useful to understand the hydrological behaviour in the watershed but also for characterizing the distribution of response times of groundwater discharge. This can be particularly useful for studying biogeochemical hotspots and hot moments (Krause et al., 2017; Singh et al., 2019; Trauth and Fleckenstein, 2017) or to couple this approach with natural tracers to assess the residence times in the hyporheic zone (Biehler et al., 2020; Liao et al., 2021)."

➢ *L. 147 « Due to some obstacles (coarse gravels, cobbles, gauging stations, etc.), it was not possible to bury the FO cable in few places. » - Don't you think that the soil structure of the riverbed*

*coresponds/correlates to the infiltrating behavior. (riverbed of a losing stream is harder than of a gaining stream). This would introduce a bias in your measurements$*

As mentioned in the discussion section, we think that the topography is the main driver of GW discharge at the watershed scale. However, it seems that local heterogeneities due to uneven bedrock weathering or to the presence of fractures can influence the nature of the streambed as well as the spatial variability of groundwater discharge.

Thus, it is true that the difficulty of burying the cable in some areas limits the possibility of fully characterizing exchanges processes since it probably prevents highlighting local heterogeneities. However, the number of measurements is high enough to characterize the general trend at the catchment scale and also local heterogeneities in many spots, which already is a very promising achievement.

➤ *L. 162 « absolute uncertainty of measurement » - Be more precise in your formulation (e.g., the standard error between DTS and RBR)*

This was included in the revised version of the manuscript: "Comparison between DTS measurements and RBR SoloT probes validated the temperature measurements, with a relative uncertainty of measurements **(standard error)** estimated at 0.05°C" (L. 181)

➤ *L. 175 « Thermal anomalies can also be identified using an analysis of the standard deviation of temperature for a given period (Sebok et al., 2015). Indeed, the calculation of the standard deviation provides insights about the thermal inertia mostly linked to groundwater fluxes towards the surface and can therefore be used to highlight relative variations of fluxes along the cable. »*
*I don't follow you here*

This sentence was removed and replaced by:

"Groundwater inflows can be detected by localizing temperature anomalies. Those can be easily identified by plotting the evolution of the temperature over time and space, especially for long time series. Thermal anomalies can also be identified using an analysis of the standard deviation (SD) of temperature for a given period (Sebok et al., 2015), since the calculation of the standard deviation provides insights about amplitudes of temperature variations. In case of groundwater discharge, the value of the SD of streambed temperature is expected to be much lower than the value of the SD of the stream temperature. Therefore, relative variations of fluxes along the cable could be determined from relative variations of SD." (From L. 192 to 197)

We hope this makes things clearer.

➤ *L.182 - Here, I would like to see the measurement and the fitted response*

Here, we made the choice to not include in the main manuscript the plots resulting from the interpretation of passive-DTS measurements. We considered that the length of the paper was in itself quite long enough and that these plots are not essential for the understanding of the data interpretation. Moreover, the plot is very classical for the method applied and does not present any specific interest. However, as detailed L.212, all the results concerning the interpretation of passive-DTS measurements with the FLUX-BOT model are provided as supplement.

➤ *L. 185 « steady over time » - Be more specific in what is steady. You are using a timeseries after all.*

It is true that the paragraph was not specific enough. We improved it by explaining more clearly the choice of the lower boundary condition:

> "To apply the model, the stream temperature and the groundwater temperature were chosen as upper and lower boundary conditions respectively. The stream temperature was measured for the wetland area at the Kerrien spring with a temperature sensor and at the gauging station E30 (RBR SoloT) for the wood plain area. The temperature signal recorded at 15-m depth in the piezometer F4 (Fig. 2b) was used to set the groundwater temperature." (From L. 205 to 208)

Concerning the term "steady over time", we removed it. Indeed, the choice of the boundary conditions is now clearly discussed in the revised version of the manuscript. A discussion about the choice of the temperature signal measured in the piezometer F4 as proxy of the groundwater temperature is now fully discussed from L. 543 to 547, which was not proposed in the initial version of the manuscript:

> "Moreover, the temperature signal recorded in the piezometer F4 was used to set the lower boundary condition of the model. This piezometer seems to be a good proxy for groundwater temperature since the temperature signal measured at 20 m-depth in the piezometer F5b is similar to the one measured at 15 m-depth in the piezometer F4. This also suggests that the temperature of groundwater discharging into the stream is uniform along the investigated portion of the catchment"

➤ *L. 197 "Concerning the thermal properties of the saturated sediments, the volumetric heat capacity was set at 3x106 J.m-3.K-1 and various values of thermal conductivities, ranging between 0.9 and 4 W.m-1.K-1, were tested."- Please elaborate on estimating the thermal properties. Homogeneous? Full calibration? In your introduction you describe the different soil types of the stream bed. As you assume that heat transport through the aquifer is the main heat transport component, these thermal properties represent an average of the aquifer*

Here, it seems difficult to estimate the value of thermal conductivities because the nature of the streambed is relatively variable in space. Moreover, as mentioned further in the study, the calibration of this parameters from passive-DTS measurements or VTPs is neither possible. That is why we propose here to test a relatively large range of thermal conductivities.

This point was probably not clear in the initial version of the manuscript since the issue of the thermal properties estimate was not addressed from the introduction. In the revised manuscript, **the introduction was improved and the choice of testing a large range of thermal conductivities values is therefore now fully justified**. Moreover, to be sure to clarify the text, we now precise from L. 217 to 220: "The streambed sediments being composed of saturated clay, silt, sand and gravel, the thermal conductivity may typically range between 0.9 and 4 W.m$^{-1}$.K$^{-1}$ (Stauffer et al., 2013). Thus, considering the importance of the sediment thermal conductivity on groundwater fluxes estimates (Sebok and Müller, 2019), the model was applied for 3 values of thermal conductivity (1, 2.5 and 4 W.m$^{-1}$.K$^{-1}$)."

- ***2.3 Active DTS measurements***

➤ *L. 199 - Maybe a section introduction?*
  *- L. 201 to 204 - You start your active section by talking about passive?*
  *- L. 205 to 210 - Shouldn't this be part of the introduction?*

Considering these 3 comments, we rewritten and rearranged the first paragraph of this section (From L. 222 to 227in the revised manuscript):

"Active-DTS measurements were conducted in April 2016 concurrently with passive-DTS measurements by deploying an additional FO cable within the streambed in the wetland, as shown in Fig. 1. While the active-DTS experiment was conducted, passive-DTS measurements had already been collected for three months, which allowed highlighting clear and significant temperature anomalies along the cable deployed in the wetland area (See results section below). Assuming that these temperature anomalies could be associated to potential groundwater exfiltration zones, the choice was made to conduct the active-DTS experiment in this area."

➢ *Fig 3 - In the other setup you are using a ambient bath and a fridge?*

Yes, that's right. For the passive-DTS experiment, the calibration was made with one cold calibration bath (fridge) and one calibration bath at the ambient temperature. In active-DTS measurements, we used a warm calibration bath instead of an ambient-temperature calibration bath. Thus, resistors were used to heat the water of the calibration bath and air pumps were set to homogenize temperature within. Using a warm calibration bath during the active-DTS experiments was essential since the bath temperatures should preferably bracket the full range of temperatures expected to occur along the cable (van de Giesen et al., 2012).

However, we didn't mention this in the initial version of the manuscript, which was a unfortunate oversight. This was corrected in the revised version of the manuscript (From L. 243 To 249):

"The calibration process applied was almost similar to the one applied for calibrating passive-DTS measurements. The only difference is that a warm calibration bath was used as reference section during the active-DTS experiment while a bath at ambient temperature was used for the passive-DTS experiment. To do so, heating resistors were set in the bath and air pumps were used to homogenize the temperature within. Considering the important temperature rise expected during the active experiment, using a warmed calibration bath is essential because the bath temperatures must preferably bracket the full range of temperatures expected to occur along the cable (van de Giesen et al., 2012)."

- *3 Results*

➢ *L. 266 – « widely attenuated » - Please quantify*

We now precise L. 308 to 311: "…temperature is relatively stable over time (around 12.5°C) and the daily stream temperature fluctuations are widely attenuated (the SD varies between 0.19 and 0.93°C while the SD of the stream temperature is 1.38°C)."

➢ *L. 306 « temporal dynamic of groundwater discharge » - Be more specific*
   *L. 306 – P1, P2, P3 not marked in 5*

These comments concern the section "Temporal variability of temperature signals". As mentioned in the general response addressed to Miriam Coenders-Gerrits, we decided to remove the section in the revised version of the manuscript in order to focus on the GW fluxes quantification.

➢ *L. 351 m « the mean flux is estimated -3.43x10-6 m.s-1 » - It is unclear to me whether this is the specific discharge from your 1D heat transport model, the velocity/seepage rate (but requires you to*

*make an assumption on porosity), or a river flow. I would expect a unit of cubic meter per second per meter river.*

Both interpretation tools (the FLUX-BOT model for interpreting passive-DTS measurements and the ADTS Toolbox for interpreting active-DTS measurements) provide estimates of the specific discharge in the z direction (i.e. the Darcy flux).

This is now clarified L. 200-201 for the interpretation of passive-DTS measurements ["This 1D model allows calculating the **specific discharge in the z direction (i.e. the vertical Darcy flux)** by inversing measured time series observed at least at three different depths."] and L. 274 to276 for the interpretation of active-DTS measurements ["The ADTS Toolbox contains several MATLAB codes that allow estimating the thermal conductivities and the groundwater fluxes (specific discharge) …"]

➢ *L. 357 « The inflows are estimated twice higher near the spring than downstream » - Twice the SD means half the flow? This assumptions requires much more explanation. One implicit assumption is that the temperature of the river averaged over the measurement period is equal along the river, and so is the temperature of the entering water averaged over the measurement period. The latter does not hold. I would expect the temperature of the groundwater entering the river averaged of the measurement period to vary along the river, due to the measurements not covering a full year and spatial variation of the residence times. The bias introduced by this difference in mean temperature becomes part of this ratio of flows*

Here, we simply wrote that "The increase of SD from upstream to downstream seems associated to a decrease of groundwater discharge. The inflows are estimated twice higher near the spring than downstream."

Like you, we don't think that the assumption "twice the SD means half the flow" is right. Indeed, there is no linear relationship between the temperature amplitude and the flow (Hatch et al. 2006). However, it is true that the results presented in Fig. 6a and b can give this impression of a linear correlation since the mean flux is estimated $-1.24 \times 10^{-5}$ m.s$^{-1}$ at d = 0 m and $-6.55 \times 10^{-6}$ m.s$^{-1}$ for d = 150 m while the value of the SD increases from 0.51 °C to 1 °C along the same interval (from 0 to 150 m).

➢ *L. 374 - You are applying a model that assumes all thermal properties are homogeneous and with the background temperature at a large distance from the fiber. Is the depth at which you burried the fiber sufficient to not be affected by this? And for sections with little water entering the river*

Concerning the thermal properties, our previous work (Simon et al. 2021) showed that the heat capacity has very little effect on the thermal response. However, during the interpretation of active-DTS measurements, thermal conductivities are not assumed homogeneous (because we previously demonstrated the importance of this parameter). Thus, the model used to interpret Active-DTS measurements (the ADTS Toolbox) provides an independent estimate of the thermal conductivity and of the groundwater flux. This point is clearly explained in the section "Materiel and methods".

Concerning the burial depth, as mentioned previously, we highlight improved the discussion about the uncertainties and boundaries conditions in the revised version of the manuscript, as suggested. (Please, see the response of your first major concern above for more details).

➢ *L. 377 - Summarize your selection procedure here*

Here, we propose to summarize the derivative method as following (From L. 267 to 270):

"Then, the data processing method developed in Simon et al. (2020) which consists in calculating the derivative of the temperature with respect to distance was applied. It allows highlighting areas where the temperature changes occur at a scale smaller than the spatial resolution of measurements, which leads to identify measurements that are representative of the effective temperature."

➢ *L. 385 « Despite some areas without data, due to noise in the data or imperfect burying, » - This is something I would be very interested in. What noise are you talking about (DTS noise would gradually change along the fiber)? How do you identify locations where the fiber is buried imperfect?*

It was probably not very appropriate to speak about the data noise here to justify the lack of data. In fact, areas without data correspond to i) spots where the cable is not buried in the sediments and ii) measurements that are not representative of the effective temperature according to Simon et al., 2020. This was not clearly explained in the main manuscript but this is entirely detailed in the supplement material.

This point was improved in the revised version of the manuscript and we hope this change will answer your questions:

" Before interpreting the data, data were processed to remove the measurement points corresponding to sections where the cable was not buried in the streambed but laid in the bottom of the stream. These sections were precisely marked during the cable installation. Moreover, since the temperature increase is mainly controlled by convection in the stream, thermal responses measured during the heating phase in non-buried sections of cable are different from thermal responses observed in buried section and are easily identifiable with temperature rises reaching steady-state in one or two minutes (Read et al., 2014). Then, the data processing method developed in Simon et al. (2020) which consists in calculating the derivative of the temperature with respect to distance was applied. It allows highlighting areas where the temperature changes occur at a scale smaller than the spatial resolution of measurements, which leads to identify measurements that are representative of the effective temperature.

Finally, among all the section used for active-DTS measurements, 172 measurements points are thought to be significant. Note that the raw data of active-DTS measurements, the data processing (sorting and quality check) and the definition of significant points are presented in detail in the supplement material." (From L.262 to 273)

➢ *L. 389 - The temperature of the is increased from 3 to > 20. In this temperature range the viscosity of water is changing significantly. Are you sure you can safely neglect the effects of a changing viscosity w.r.t. its surroundings?*

Indeed, the viscosity and the density of the water are temperature-dependent. We made a fast calculation to try to evaluate this effect. At 12°C, the ratio density/viscosity is around 809, while this ratio is around 1382 at 35°C, meaning the hydraulic conductivity is 1.7 times higher at 35°C.

However, it seems important to take into account the following points. First, during the heating period, the temperature measured is the temperature of the FO cable, and not the temperature of the water surrounding it. The temperature signal includes the heat stored within the cable itself, which represents an important part of the increase (estimated at 14.06°C here considering the high heating rate). Thus, the water temperature does not increase as much as it seems in first glance. Thus, the effect of temperature on viscosity and density is less significant: the hydraulic conductivity is only 1.2 times higher when the temperature varies between 12 and 21 °C for instance.

Moreover, we recall that the temperature increase is very localized around the heated FO cable and decrease very rapidly away of the cable.

➤ *L. 390 « (Simon and Bour, Submitted) » - Already cited multiple times before*
We removed this citation. The paper is now cited only 3 times in the paper.

- **4 Discussion**

➤ *L. 483 « which seems technically impossible » - why ?*

Burying three FO cables at three different depths in the streambed is more than challenging. At least, it cannot be done manually. It could be done by any company specialized in the installation of submarine cables or in laying cables beneath rivers but it would certainly highly alter the streambed which raises environmental and ecological concerns. Moreover, the uncertainty linked to the burial depth of each cable would be a major limitation to estimate groundwater fluxes.

➤ *L. 493 « If stream temperature variations are assumed almost uniform along the studied section, the values of SDs inferior to the SD of the stream temperature could actually be interpreted as groundwater inflows. » - If the temperature is uniform then the SD is also the same everywhere. Please be more specific*

This point was clarified in the revised version of the manuscript (From L. 488 to 502):

> "Interpreting the data without measuring stream temperature variations along the stream requires assuming that the stream temperature is uniform along the investigated section. In this case, the value of SD of the stream temperature can be assumed uniform as well along the investigated section and any value of SD measured in the streambed lower to the SD of the stream temperature could actually be interpreted as the result of groundwater inflows."

➤ *L. 560 « The results suggest a decrease of the groundwater discharge from upstream to downstream, with the most significant inflows located in the first 20-m of the heated section. Elsewhere, groundwater discharge is more diffuse in space. Results also highlight very punctual increases of groundwater discharge matching with punctual decreases of streambed temperature SD values » - This is a misleading statement. I would say that at more than half of the locations the estimates differ a factor two.*

We propose to change this sentence for a better understanding:
> "Elsewhere, groundwater discharge is more diffuse in space although significant groundwater discharge areas can be locally observed. These local increases of groundwater discharge match with isolated decreases of streambed temperature SD values, calculated from passive-DTS measurements" (From L. 597 to 599)

➤ *L. 563 « Thus, active-DTS measurements allow validating the hypothesis made to interpret passive-DTS measurements (diffuse inflows with very punctual increases of groundwater discharge). » - I do not agree to this conclusion*

This sentence was not fully relevant. Indeed, active-DTS measurements do not allow directly validating the hypothesis made to interpret passive-DTS measurements but is in very good agreement with the general behavior/trend highlighted through passive-DTS measurements. It suggests that the hypothesis on which the data interpretation of passive-DTS measurements are based on are reasonable

(otherwise results would have been different). But , to avoid any misunderstanding, the second sentence was removed from the revised paper and replaced by :

> "Of course, results of active-DTS measurements are useful to validate the general behaviour/trend highlighted through passive-DTS measurements, that is a baseline of groundwater discharge associated to local and important spikes of discharge. They do not fully allow validating the different hypothesis made to interpret passive-DTS measurements but they permit to check the consistency of the results obtained." (From L. 630 to 634).

Note that this paragraph was moved in the discussion section concerning the complementarity of both approaches.

➢ *L. 579-580 - I was expecting an estimate of the minimum depth the cable needed to be buried in order to still be able to assume homogeneous soil around the cable*

The minimum depth to bury the cable depends on flow direction (it should be deeper in gaining conditions) and on its intensity. As mentioned in the paper, it would clearly require complementary tests to fully describe the effect of the near river on the thermal response, which goes much beyond the aim of the manuscript..

➢ *L. 588 « the length of the heated section is limited, due to the electrical injection limitation » - Please use correct wording*

This was corrected (From L. 524 to 525): "Moreover, this method requires more instrumentation and the length of the heated section is limited depending on the power supply available"

**_Response to the comments of Stefan Ploum (Reviewer 2)_**

*This study investigated groundwater-surface water exchanges along a headwater system, through the use of both passive and active thermal methods. The goal of this study was to quantify spatiotemporal variability in groundwater discharges to streams, which is an important and timely challenge. The authors have presented a novel combination of field data, which consisted of both active and passive use of FO-DTS (fiber-optic distributed temperature sensing), vertical thermal profiles, and hydrometric data. While this study seems to fill an important knowledge gap in the quantification of groundwater-surface water exchanges, the manuscript is for a large part focused on relative differences and qualitative evaluation of the observed thermal gradients. Similar to Reviewer 1, I suggest to develop the parts regarding quantification of GW fluxes/seepage rates. As such, I consider this study appropriate for HESS audience and to be publishable after considerable revisions.*

*In general, the manuscript is well structured, and reads pleasantly. The language is descriptive and engaging, which helps to tag along with the implications and meaning of the observations that are presented. I would recommend to more clearly provide a research question and/or hypothesis to understand what is tested and what the concrete outcomes are. In the current state I have the impression most of the manuscript is interpretation of observations, which are then coupled to potential explanatory factors. As such, the manuscript leans heavily on relative comparison of standard deviations of thermal records and interpretation thereof. While this is interesting and helps to understand the system that is studied, the novelty of the manuscript lies in the combination of active and passive DTS measurements and the eventual goal to quantify groundwater fluxes. The latter unfortunately has been covered quite sparsely. Therefore, I would recommend to compress the interpretation of relative differences in temperature/SD, and leverage the quantification of fluxes such as in Figure 9, which I found personally the most compelling aspect of this paper. Also the approximation in section 4.2 of relative GW contributions to total discharge could be further expanded (L565), and could make this study more enticing. However, to develop the quantitative aspect I believe that the underlying assumptions (e.g. 1D model of thermal gradient and homogenous stream T) need to be part of this expansion. Further, the discussion is for a large part evaluating methodological limitations, which are of interest but leave little space for a more broad interpretation of the results and contextualizing the presented work. In the discussion it would be interesting to read more on the relationship between spatial distribution of soil/sediment, thermal conductivity, hydraulic properties and implications for the DTS observations. Also linking this heterogeneity in and around to stream channel to heterogeneity of the landscape and riparian zone could be an interesting topic of discussion. While I think that the first part of section 4.3 was going in this direction and could be further expanded and complemented with e.g. hydrological connectivity (for example Jencso et al. 2009, doi:10.1029/2008WR007225)*

Dear Stefan Ploum,

Thank you for your time and for reviewing our work. We are pleased to read your interest for our study and for the developed approach. Through your comments, it clearly appears that we did not sufficiently manage to highlight the outcomes, the perspectives and the full potential of the described approach.

Thus, with this in mind and following your comments, we significantly revised the manuscript in order to clearly highlight the main objective of this work that is the quantification of groundwater discharge in headwater catchments. To do so, the following revisions of the manuscript:

- **The abstract was fully rewritten**. The aim was to refocus the text on the research question that is the fluxes quantification and to propose a more compelling abstract as you suggested. We now address the difficulty of quantifying GW discharge and results are more clearly presented.
- **The introduction was strongly revised** to better highlight our objectives.
- All along the manuscript, we will try to shorten the text and especially the section concerning the analysis of the standard deviation of temperature. Thus, we **removed the section 3.2** "Temporal variability of temperature signals" that was indeed not fully relevant **given the research question**.
- When presenting the results, we carefully address the issue about uncertainty estimates, both for passive-DTS and active-DTS measurements.
- Last but not least, we will also **fully rearrange the discussion section in order to focus on the understanding of groundwater discharge processes**. We reorganized it according to the main outcomes. In the revised version, we now first focus on the detection and the localization of GW discharge areas in order to highlight the usefulness of passive-DTS measurements that can be used to efficiently detect groundwater discharge at high resolution under very long section of FO cable and to monitor temporal variability of discharge. Then, the discussion focuses on the quantification of GW discharge. For this point, we show that passive-DTS measurements are not sufficient since the uncertainties on fluxes estimates are too high (as suggested by the reviewer 1, we will strengthen the question of uncertainties and boundaries conditions). We demonstrate here that conducting active-DTS measurements is a very promising approach to precisely quantify GW discharge into streams. The resolution of estimates is particularly interesting to characterize spatial variabilities at small scale. Such results cannot be achieved with any other methods commonly used in this context. Then, to go further and as suggested, we highly improved on the section concerning the complementarity of both methods to quantify GW discharge. This point is now clearly presented as a major result.

Then, the section 4.3 (called "Understanding of groundwater discharge dynamics") was also revised in order **to discuss the interest of the approach to characterize GW discharge and its relationship with catchment structure and properties**. We focused on the fact that the approach allows characterizing GW discharge at two scales. First, we discuss the correlation between the catchment topography and GW discharges that seems demonstrated here before interesting in the possibility of highlighting small scales spatial heterogeneities. Of course, this discussion remains quite limited since we have few knowledge about the soil properties, soil conductivities…

Lastly, your comments bring us in mind that we didn't talk enough out the promising perspectives that offer the method. Indeed, being able of precisely quantifying GW discharge at small scale open new windows to study biogeochemical hotspots and hot moments, to characterize response times or to assess the residence times of water in the hyporheic zone.

We join with this submission a document containing all changes made. We hope these revisions will answer your main concerns. Once again, we thank you for your comments that should allow highly improving our manuscript.

Please find following a point-by-point response to all technical details and text edits you addressed to us.

Kind regards,

Nataline Simon & co-authors
* * *
*Further, I like to address a number of technical details and/or text edits. Below I have provided some by line number, but they might apply to the entire manuscript (e.g. consistent use of units or language). While the paper reads pleasantly, there are some consistent language errors that are a little bit distracting or could be confusing. However, as I am not a native speaker, take my suggestions with that in mind and maybe double check with others.*

➢ *L11, L15 and further: "consist of"*

After checking, we think that the use of "consist in" in the text are appropriated.

➢ *L14 and further: unsure of the use of punctual here and elsewhere. I suggest point measurement but this depends on the use.*

Indeed, the use of « punctual » is not appropriate here. This is now changed in the revised manuscript all along the text (corrections were adapted according to each context).

➢ *L23: On the opposite -> contrary*

This was changed in the revised version of the manuscript (and each time it was necessary).

➢ *L25: the end of the abstract dies down a bit, is there a more compelling way to conclude this work?*

As explained above in the response to your main concerns and in the general response to the editor, we fully rewrite the abstract and we hope the end of the abstract is more compelling henceforth.

➢ *L90: as mentioned I would suggest a more concrete research question. The aim to discuss a certain topic feels like a misfit with the data that is available and the eventual goal to quantify fluxes.*

Also as explained previously, we propose in the revised version of our manuscript to focus on the detection and the quantification of fluxes (instead of focusing on the methodology). Thus, this last paragraph of the introduction was rewritten in order to redefine the research question and the objectives of the study (From L. 99 to 111):

> "In this study, we propose to use for the first time active-DTS measurements to quantify groundwater discharge in the stream of a headwater catchment. The application of active-DTS methods in such context is particularly promising since the interpretation of active-DTS measurements in saturated porous media provides estimates of both sediments thermal conductivities and groundwater fluxes over a large range and with an excellent accuracy (Simon et al., 2021)., This method should allow quantifying groundwater discharge and characterizing the streambed thermal properties at an unprecedent spatial scale. In complement of active-DTS measurements, which were limited in space and time, a long-term passive-DTS experiment was conducted at the catchment scale in order to infer the location and dynamics of groundwater discharge. Therefore, this study also investigates how these two experiments could be compared and combined to characterize both the spatial and the temporal dynamics of groundwater discharge. To do so, FO cables were deployed in the streambed sediments of a headwater stream within a small agricultural watershed. In the following, we first present the headwater

watershed and the experimental setup before presenting the methods used to interpret both passive- and active-DTS measurements. Fluxes estimates obtained with both passive- and active-DTS measurements are then compared and the advantages and limitations of each method are finally discussed."

➢ *L92: For doing so => To do so*

This was modified in the revised version of the manuscript (each time it appeared in the initial version of the manuscript).

➢ *L100: achieved on => conducted in*

This was modified in the revised version of the manuscript.

➢ *L104: researches => research*

This was modified in the revised version of the manuscript.

➢ *L104: consider revising the sentence to something like: "This site was selected because it had the advantage of readily installed equipment, for monitoring and experimental studies (Fovet et al 2018)."*

The sentence L.104 was revised to: "The site is a part of the French network of critical zone observatories (Gaillardet et al., 2018) and supports extensive hydrological and geochemical research. This site was selected because it presents the advantage of readily installed equipment and instruments (Fovet et al., 2018)." (L. 117-120 in the revised manuscript)

➢ *L108: higher slopes => steeper slopes*

This was modified in the revised version of the manuscript.

➢ *L109: was the wetland developed my people? If natural, I would suggest to change was developed to e.g. "is positioned" or "is situated"*

This was changed to: "characterized by a large wetland" (L. 126 in the revised manuscript)

➢ *L110: This is a man-made environment where... => In this man-made/anthropogenic environment the stream…*

This was modified in the revised version of the manuscript.

➢ *L112-L114: This made me wonder what type of drains these are? I would argue that rain water that is quickly routed through the shallow soil layers into belowground drains could still be considered as a form of groundwater given that it chemically and/or thermally has been affected by this short residence time belowground. In the light of spatial heterogeneity of GW-stream interactions this might be an important nuance (e.g. Hester and Fox 2020, doi/10.1029/2020WR028186)*

Such would modify strongly the hydrological cycle and groundwater/surface water interactions. Maybe it could explain why no significant groundwater discharge is observed in this part of the watershed. However, they are other parts of the basin where there are no drains and where no

significant groundwater discharges are observed. Thus, although this might affect the watershed hydrology, it is difficult to make a clear conclusion about it. Moreover, our main results come from the wetland area where there are no drains.

> *L123: precipitation is*

This was modified in the revised version of the manuscript.

> *L124: Reference for previous studies, and "on average"*

This was modified in the revised version of the manuscript.

> *Figure 2: suggest to change line types for colorblind people. This could be good to improve in other figures too, but see where you find the opportunity.*

We tried to change the solid lines into dashed or dotted lines but the result was unfortunately not really convincing (it does not avoid using several colors to distinguish the different curves). Thus, although we tried to favorably respond to this request, it was not possible to obtain a satisfactory result.

> *Figure 2b: the lag between thermal signals would almost suggest to me that there is recharge rather than discharge in the riparian zone. I think Reviewer 1 has expanded on this more, but I think it should in general be clear whether gaining or loosing conditions apply, and what the implications are for the GW fluxes and the methods used. Especially since the 1D approach does not provide a clear direction but only rates, this can be an important issue.*

Indeed, Reviewer 1 also questioned whether gaining or loosing conditions apply in the riparian zone. In fact, there was a misunderstanding, which was due to a confusion in the legend of the Fig 2a. We improved the legend of the Fig which clarifies things: F2 is near the stream, F1b higher in the catchment and the piezometric level measured in F2 is > than the one measured in F1b which confirms gaining conditions in the riparian zone. This result was also confirmed through the interpretation of passive-DTS measurements.

Concerning thermal signals showed in Fig. 2b, it is true that it could also suggest losing conditions because of temperature signals recorded in the piezometers F2 and F1b. However, the lags observed between the signal measured in the stream and the ones measured in piezometer are probably due to the diffusion of air temperature variations through the water columns of piezometers. To verify this assumption, we used a finite-difference model to solve the 1D heat equation (Anderson, 2005). We solved here the 1D heat equation considering heat conduction through the water column but no advection (no vertical flux in the piezometer; q=0) (Anderson, 2005). The thermal conduction coefficient (or diffusion coefficient) was set at $1.41 \times 10^{-7}$ m².s$^{-1}$. This simple model allows estimating the "diffusion time", i.e. the time required for air temperature variations to propagate by conduction in the water column. Results show that approximatively 70 days are required so that air temperature variations reach 2 m-depth and approximatively 170 days for 5 m-depth. These results are fully consistent with time lags observed in F2 and F1b (Fig. 2b). This confirms that temperature variations recorded in these piezometers result from the diffusion of air temperature variations through water columns of piezometers and not from the losing conditions in the riparian zone. To make things clear, the following sentence was added in the revised manuscript :

"While the groundwater temperature is almost constant in the upslope domain (piezometers F5b and F4), temperature variations recorded in the stream and in the downslope domain (F2 and F1b) show larger variations following daily and seasonal temperature variations. It can easily be shown that temperature variations recorded in F2 and F1b result from the diffusion of air temperature variations through the water columns of piezometers." (From L. 151 to 154)

➢ *L145: achieved => conducted*

This was modified in the revised version of the manuscript.

➢ *L148: suggest to change to "the average burial depth was estimated to be 8 cm"*

This was modified in the revised version of the manuscript.

➢ *L152: thicker => deeper? I think this is personal preference*

This was modified in the revised version of the manuscript.

➢ *L165: Are there at the vertical profiles also coils of DTS cable that allow some form of comparison?*

No there aren't. In fact, VTP were installed way after the passive-DTS cable. This is why the precise comparison of both results is tough. Only a rough approximation of the location of VTPs compared to DTS measurements is achievable here.

➢ *L172: Suggest to start with: "Groundwater inflows can be detected by…*

This was modified in the revised version of the manuscript.

➢ *L187: the assumption of the homogenous stream temperature applies to only at the VTP? To me this sounds like you assume homogenous temperatures along the reach, which for obvious reasons would be a quite bold assumption.*

In fact, the interpretation of passive-DTS measurements also relies on the assumption that the stream temperature is homogeneous all along the investigated method. Here, this is the only way to quantify fluxes from passive-DTS measurements and to validate the qualitative interpretation (lower values of SD, that are thermal anomalies, associated to preferential GW discharge areas). However, as suggesting by reviewer 1, the question of uncertainties associated to boundaries conditions was largely improved in the revised version of the manuscript. We now discuss from L. 496 to 509 and from L. 538 to 548 assumptions of an homogeneous stream temperature, its consistence and its consequence on fluxes estimates uncertainties. Please see the major comments of Reviewer 1 for more details.

➢ *L201: this reads as results and therefore comes as a surprise. Maybe consider rephrasing or remove the first lines.*

It is true that the choice of conducting active-DTS measurements in the riparian zone was made following the first results obtained with passive-DTS measurements. We tried to rephrase this paragraph in order to better explain this :

" Active-DTS measurements were conducted in April 2016 concurrently with passive-DTS measurements by deploying an additional FO cable within the streambed in the wetland, as shown in Fig. 1. While the active-DTS experiment was conducted, passive-DTS measurements had already been collected for three months, which allowed highlighting clear and significant temperature anomalies along the cable deployed in the wetland area (See results section below). Assuming that these temperature anomalies could be associated to potential groundwater exfiltration zones, the choice was made to conduct the active-DTS experiment in this area." (From L. 222 to 227 in the revised manuscript)

➤ *L231: I think the data interpretation section can be merged with the previous one.*

Done. Thus, the subtitles 2.3.1 and 2.3.2 were removed

➤ *Figure 4: is the red line indication an average stream temperature of 1.38 not a point (around 500 meter if I'm correct?)*

The red line indicates the average SD of stream temperature, which was calculated using temperature measurements recorded at the gauging station E30, as mentioned in the caption of the Figure and in the text.

➤ *L270: local spikes or dips or anomalies? Peaks would imply that it always goes up*

You are right that the term "peak" is maybe not well appropriate here. It was changed in the revised manuscript by the term "spikes" (as many times as necessary in the text)

➤ *L271: associated with (also line 273), very run on sentence*

This was modified in the revised version of the manuscript.

➤ *Figure 5: in the text panel b is discussed first, then panel a. Maybe take this into account when revising*

In fact, Fig 5a is discussed before Fig 5b in the manuscript.

➤ *L338-L340: "values of" can be omitted*

Done

➤ *L341: d implies depth but refers the distance? Maybe consider length (l)?*

Here, d refers to the distance (along the FO cable).

➤ *L355: "Regardless of" might be more suitable than "Whatever the uncertainties"*

This was modified in the revised version of the manuscript.

➤ *L364: the experiment*

This was modified in the revised version of the manuscript.

➢ *Section 3.2 and elsewhere: I suggest to express the GW fluxes in m/d since in m/s values are so low it is difficult to imagine*

We prefer here keeping the GW fluxes in m/s. We are aware the values are very low but we decide to use the International system of units.

➢ *Figure 10: maybe consider splitting up the figure, since there is a lot going on.*

It is right that there is a lot going on this Fig. However, this plot aims to compare the results of fluxes estimates obtained with the different approaches. Thus, we think that having a single plot remains the most efficient way to reach this goal.

➢ *L480: rephrase "conclude about"*

Done. This was changed in the revised manuscript to: "discuss"

➢ *L482: not sure if I follow the thought about needing 3 DTS cables, this seems like an practical issue that is relevant, but maybe I'm missing something*

Deploying 3 FO cables at different depths would really improve the GW quantification, as demonstrated by Mamer and Lowry (2013). With a single FO cable, we have to make assumptions about boundaries conditions in order to interpret GW fluxes (here we assume for instance GW temperature constant over time and stream temperature uniform all along the transect). Thus, collecting distributed temperature measurements at three different depths allows highly reducing the uncertainties while quantifying vertical fluxes since temperature at upper and lower boundaries is measured (and not assumed or extrapolated).

To make things clearer, some explications were added in the text: "Ideally, the approach would require deploying at least 3 FO cables in the field (Mamer and Lowry, 2013) in order to continuously measure temperature conditions at high resolution at upper and lower boundaries, which was technically impossible in the field." (L. 540 – 542 in the revised manuscript)

➢ *L563: which hypothesis?*

This sentence was not fully relevant. Indeed, active-DTS measurements do not allow directly validating the hypothesis made to interpret passive-DTS measurements but allow validating the general behavior/trend highlighting through passive-DTS measurements. It suggests that the hypothesis on which the data interpretation of passive-DTS measurements are based on are correct (otherwise results would have been different). But these assumptions are not validated for all that. Thus, the second sentence was removed from the revised paper and replaced by :

"Of course, results of active-DTS measurements are useful to validate the general behaviour/trend highlighted through passive-DTS measurements, that is a baseline of groundwater discharge associated to local and important spikes of discharge. They do not fully allow validating the different hypothesis made to interpret passive-DTS measurements but they permit to check the consistency of the results obtained." (From L. 630 to 634).
Note that this paragraph was moved in the discussion section concerning the complementarity of both approaches.

➢ *SI: L69: section 3 header says passive which should be active?*

You are right, this was an error, which was corrected in the revised version of the supplement material.

➢ *Figure S4, panels b and c: Does power outage refers to termination of the experiment? Now it reads as a cut-off of the power supply, but I assume it is the moment the heat was turned off.*

Yes indeed, the term "power outrage" refers to the end of the experiment. It is now specified in the caption of the Fig. that "The term "Power outage" marks the end of the heating period."

---

## Referee Report (RR1)

[revised manuscript text omitted]
 2x10$^{-6}$ and 4.74x10$^{-5}$ m.s$^{-1}$, with a mean value at 1.34 x10$^{-5}$ m.s$^{-1}$ and a SD of 9.18 x10$^{-6}$ m.s$^{-1}$. For 9 locations (blue points), only the value of $q_{lim}$ was evaluated since the departure of the conduction regime towards temperature stabilization was not reached at the end of the heating period. Note that the data interpretation does not provide the flow direction, the temperature increase being identical for upward and downward conditions. Although significant measurements are not available all along the sections, results show a decrease of the flux from upstream to downstream, particularly in the first twenty meters of measurements. At greater distances, fluxes are more diffuse in space, except at few locations, for instance at 43, 50 and 52 m from the start of the heated section where higher values are observed. Interestingly, very local and high fluxes values, spreading on less than 2 m, can be observed, as for instance at $d = 10$ m.

[Figure]

550

**Figure 89: The interpretation of active-DTS measurements along the heated section of FO cable leads to estimate the spatial distribution of both a. the thermal conductivity (uncertainty = ± 0.2 W.m$^{-1}$.K$^{-1}$) and b. the water fluxes and their associated errors (error bars). Blue points mark locations where the temperature stabilization is not reached and where an estimate of $q_{lim}$ is provided. Errors bars corresponding to uncertainties on flow estimates calculated with respect to data noise for each measurement**

555 **points.**

**3.3 Comparison between passive- and active-DTS measurements**

Figure 10 9 compares estimated values of groundwater fluxes for the 7$^{th}$ April 2016. The flow direction is assumed upward in agreement with passive-DTS measurements (Fig. 7b6b).

For passive-DTS measurements, the two light grey curves correspond to fluxes estimates considering $\lambda = 1$ W.m$^{-1}$.K$^{-1}$ and $\lambda = 4$ W.m$^{-1}$.K$^{-1}$, assuming that the effective values of fluxes vary should range between these two

560 thresholdsestimates. It appears thatThe estimation of the flow quantificationgroundwater discharges clearly remains highly 
[revised manuscript text omitted]

---

## Author Response (AR2)

Dear Miriam Coenders-Gerrits,

Thank you very much for your feedback and for considering our study as a potential publication for HESS. We appreciated that the reviewer agreed to review for the second time our manuscript and we thank him for his time.

Please find below a detailed point-by-point response to reviewer's comments. We carefully answered to every single comment/question addressed and also considered the few comments he made in the attached document.

Thank you for your time,

Kind regards,

Nataline Simon & co-authors.
* * *
**Response to the comments of Bas des Tombe**

Dear Bas des Tombe,

First, we would like to thank you very much for accepting reviewing once again our manuscript and for your comments. Please find in the following, a point-by-point response to your comments. Your comments are *in italics* and the responses in regular text.

Kind regards,

Nataline Simon & co-authors

*- Remove the suggestive claims that certain dynamics of variability in seepage flow are measured with the passive method. That would require multiple seasonal cycles, while the measurement period here is 8 months.*

Indeed, the passive-DTS measurements were "only" conducted for 8 months. It is true that it is not sufficient to fully investigate the temporal pattern of GW discharge. However, it appears that these 8 months of monitoring are enough to highlight temporal changes in groundwater discharge. This is what we show in sections 3.1.1 and 3.1.2. The analysis of time temperature series (3.1.1) and the model used to quantify flow from passive-DTS measurements (3.2.1) both suggest an increase of groundwater discharge in winter (induced by the increase in precipitations that contribute to increase hydraulic gradients) and a decrease in spring (in agreement with the decrease of the groundwater table). This point is widely discussed in the discussion section (4.3). Therefore, we are convinced that we are actually able to highlight a certain dynamic in seepage flow.

However, as you mention in the attached document, the sentence "Long-term passive-DTS measurements turn out to be an efficient method to detect and locate groundwater discharge along several hundreds of meters to investigate the temporal pattern of

exchanges over the annual hydrological cycle" is probably too general and not enough precise. Thus, this sentence is now improved in the revised abstract:

". Long-term passive-DTS measurements turn out to be an efficient method to detect and locate groundwater discharge along several hundreds of meters. **The continuous eight-months monitoring allowed highlighting changes in the groundwater discharge dynamic in response to the hydrological dynamic of the headwater catchment**"

*- Claims of the authors being the first to attempt active heat tracer tests with DTS in stream/river beds to quantify flow are simply not true. https://www.mdpi.com/1424-8220/20/19/5696/htm#B65-sensors-20-05696 / https://hess.copernicus.org/articles/19/2663/2015/hess-19-2663-2015.pdf*

We are sorry to read this comment because we didn't intend to claim that we were the first to attempt active heat tracer tests with DTS in stream/river beds. Thus, the second reference you mention here (Kurth et al. 2015) is referred in the introduction (from L. 95 to 98): "Despite promising developments, active-DTS methods have been seldom used in hydrology to estimate groundwater/surface water interactions. Kurth et al. (2015) coupled passive- and active-DTS measurements and highlighted areas with lower and higher flow rates over the cable…". However, in their study, Kurth et al. (2015) do not go as far as us since they don't use active-DTS measurements to quantify fluxes. In their study, active-DTS measurements are only used qualitatively to locate lower and higher flow rates. This is why we add then in the text: "…but the quantification of fluxes remained unexplored."

However, as you mention in one of the comments in the attached files, the following sentence in the abstract: "On the contrary, active-DTS measurements, which have never been in streambed sediments up to now…" is not exactly true since active-DTS measurements have actually been conducted in streambed (but not to quantify fluxes). Thus, this sentence was improved in the revised version (L. 22 in the abstract): "On the contrary, active-DTS measurements, **which have seldom been performed in streambed sediments and never applied to quantify water fluxes** …"

Concerning the first reference you mention (Ghafoori et al. 2020), we are a little surprised about this comment. This paper deals about the use of DTS measurements to detect and quantify seepage flow through embankment dams. Such application is very different from the one proposed here. Note however that this study is already referred in the introduction section (L. 92) when we introduce previous applications of active-DTS methods.

*- Punctual measurements -> point measurements*
Term "Punctual measurements" is used in the manuscript for two different purposes.

- Firstly, the term was used twice concerning active-DTS experiment (we used the term "punctual active-DTS measurements" to indicate the experiment was performed punctually in time). In this case, this term was removed in the revised manuscript and associated sentences were rephrased. Instead of speaking about "punctual active-DTS

measurements", we now precise in the revised manuscript: "the active-DTS experiment, performed during few days **...**" (L. 16 and 285).

- Secondly, the term was used twice in the abstract concerning the Vertical Temperature Profiles. In these cases, the term "punctual measurements" was replaced by "point measurements" as suggested (L. 58 and 80 in the revised manuscript).

*- The flow rates can be so high that thermal dispersion, thus a velocity dependent diffusion term, heavily affects the flow estimates. (specific discharge >~ 0.5m/day)*
*- Thermal dispersion is still left out of the equation, even though flow velocities are high (q>~0.5m/day).*

In our case, we consider that it is not necessary to consider thermal dispersion effects since a single FO cable is used during the active experiment (no transport between the heat source and the FO cable). Incidentally, studies presenting active-DTS measurements conducted with heated FO cables – for instance Munn et al. (2020), Maldaner et al. (2018), Coleman et al. (2015) or else Bakx et al. (2019) – do not consider neither this parameter. However, while an independent heating cable is used as heat source (Bakker et al. 2015, des Tombes et al. 2018, among others), it is certain that thermal dispersion effects can affect flow estimates, especially for large values of specific discharges.

*- And the estimates of the passive test seem to be heavily dependent on the value chosen for the thermal conductivity.*

This is absolutely correct. For passive-DTS experiment, flow estimates are highly dependent on the thermal conductivity. Actually, this is one of the main conclusions of this study. We demonstrate that flow quantification from passive-DTS measurements is highly limited because of the uncertainty on thermal conductivities. This is why, we propose to assess the effect of thermal conductivity on flow estimates (for instance in sections 3.1.2 and 3.3). This point is clearly considered in the conclusions of the study and detailed in the discussion section (see the section 4.2.1):

"Last but not least, results showed that thermal conductivity values have a very strong impact on fluxes estimates, which is consistent with the results of Briggs et al. (2014), Duque et al. (2016), Lapham (1989) and Sebok and Müller (2019). The lack of knowledge and assumptions on thermal conductivities values lead to high uncertainties on fluxes estimates using both VTP and passive-DTS measurements (Fig. 6b). In-situ estimates of thermal conductivities using thermal conductivity probes could considerably improve the fluxes estimates, as demonstrated by Duque et al. (2016), who reported up to 89% increase in flux estimates when using in situ measured sediment thermal conductivities. However, seeing the high spatial variability of the thermal conductivity highlighted through the active-DTS experiment, it would certainly require a tremendous effort in the field to characterize such variability with single probes. Moreover, it will not remove others sources of uncertainties associated to the burial depth of the FO cable or to the lack of temperature measurements at different depths all along the section."

Nevertheless, we also demonstrate that coupling passive-DTS measurements with active-DTS measurements can be an efficient way to counter this limitation. This is discussed in section 4.2.3 and is one of the main conclusions of the study.

*- All measurements were performed in a river that is gaining. Both passive and passive experiments suffer more from the false boundary conditions in rivers under losing conditions. A gaining river gains water with a nearly constant temperature, in a losing river the water temperature can fluctuate wildly. The title and the scope of the presented manuscript should be limited to gaining streams. The word "into" in the title is, in my opinion, not sufficient and I would propose to add the word gaining to the title.*

We have to say that we are a little surprised with this comment, especially because you already raised this point in your previous review. At this moment, we completely agreed that this point deserved to be improved. Thus, in the revised manuscript we submitted in December, we significantly improved this.

First, we changed the title of the study. According to us, this is clearly expressed starting from the title that the study focuses on gaining conditions:

"Combining passive- and active-DTS measurements to locate and quantify **groundwater discharge** variability **into a headwater stream**"

Indeed, the title includes two terms ("groundwater discharge" and "into a headwater stream") that are both related to gaining conditions (and no confusion is possible here).

Moreover, significant changes were made to improve the abstract. Thus, the two first sentences of the abstract also clearly introduce this point: "Exchanges between groundwater and surface water play a key role for ecosystem preservation, especially in headwater catchments where **groundwater discharge into streams highly contributes to streamflow generation and maintenance**. Despite several decades of research, investigating the spatial variability of **groundwater discharge into streams** still remains challenging mainly because groundwater/surface water interactions are controlled by multi-scale processes."

Then, a bit further (4$^{th}$ sentence of the abstract), the scope of the study is explicit: "To do so, we propose to combine, for the first time, long-term passive-DTS measurements and active-DTS measurements by deploying FO cables in the streambed sediments of a **first- and second-order stream in gaining conditions**."

*- Please have a look at the few comments in the attachment*

Thank you for the comments in the attached file, which allowed improving the abstract in particular and correcting some spelling mistakes and typing errors.